# Global Development and Readiness of Nuclear Fusion Technology as the Alternative Source for Clean Energy Supply

**Mustakimah Mohamed [1], Nur Diyana Zakuan [1,2], Tengku Nur Adibah Tengku Hassan [1], Serene Sow Mun Lock [1,2] and Azmi Mohd Shariff [1,2,*]**

1   CO2 Research Centre (CO2RES), Department of Chemical Engineering, Universiti Teknologi PETRONAS (UTP), Seri Iskandar 32610, Perak, Malaysia
2   Department of Chemical Engineering, Universiti Teknologi PETRONAS (UTP), Seri Iskandar 32610, Perak, Malaysia
*   Correspondence: azmish@utp.edu.my; Tel.: +60-53687530

**Abstract:** Nuclear fusion is understood as an energy reaction that does not emit greenhouse gases, and it has been considered as a long-term source of low-carbon electricity that is favourable to curtail rapid climate change. Fusion offers a pathway to resolve energy security and the unequal distribution of energy resources since seawater is its ultimate fuel source and a few grams of fuel can generate mega kilowatts of power. The development and testing of new materials and technologies are unceasing to achieve the net fusion energy through national and international collaboration as well as private partnerships. The ever-growing number of research works report various designs and magnet-based fusion devices, such as stellarators, lasers, and tokamaks. This article provides an overview on the utilization of nuclear energy as a clean energy source, as well as the strategies and progress towards establishing successful commercial fusion energy to the grid and transition to a reliable clean energy source. The overview focuses on the fusion nuclear development in five major countries, UK, US, China, Japan, and Russia. Identified technical and financial challenges are also described at the end of this article. The International Thermonuclear Experimental Reactor (ITER) has been an international reference program for fusion energy development and most developed countries with nuclear development capacity are aiming to complete their in-house fusion energy facilities in parallel to ITER. Many fusion programs are finishing the conceptual design and shifting into the phase of engineering design for the planned DEMO fusion facilities. The significant challenges were identified from the perspective of device efficiency and robustness, sustainable funding, and facility maintenance and safety, which must be addressed diligently to realize fusion energy as alternative clean energy that mitigates climate change and supports the goals of energy security.

**Keywords:** clean energy; fission reaction; fusion reaction; ITER; fusion devices; tokamaks

## 1. Global Outlook of Nuclear Energy for Power Generation

Clean energy and energy security have been the major critical elements strived for by most countries in the world to drive their economic sustainability and living quality. Energy that is affordable, clean, stable, and sustainable are the types of energy that capture global interest for the transition to renewable energy sources and are the means to combat greenhouse gas (GHG) emissions and climate change. However, the roll out of renewable energy sources for the energy mix such as solar, wind, hydrothermal, and biomass has illustrated their challenges. These sources are not uniformly available and are vulnerable, limited, and fluctuate depending on geopolitics and climate changes. The outlook for alternative clean and sustainable energy sources has been expanded into advanced technology that requires larger investments, extensive exploration, and plausible demonstration.

Nuclear energy is one of the alternative innovative technologies that has growing interests globally for stable and clean energy generation. It offers minor GHG emissions

and has the potential to be a cost-competitive technology in a long-run operation [1]. The Intergovernmental Panel on Climate Change (IPCC) explained that nuclear power can provide stable low-carbon electricity. For instance, the UN's Economic Commission for Europe (UNECE) reported the range of $CO_2$ released from nuclear energy in 2022 was 5.1–6.4 g $CO_2$/kWh, which is the lowest $CO_2$ emitted among all power generation technologies. Figure 1 compares the number of $CO_2$-equivalent emissions per unit of electricity generated by major energy sources based on the lifecycle analysis conducted by the United Nations (UN) IPCC [2]. It indicates the median value for $CO_2$ emitted from nuclear power plants is 12 g $CO_2$/kWh, which is equivalent to the amount of $CO_2$ emitted from wind and lower compared to solar and other sources [3]. Coal and biomass co-firing are recognized as the energy sources with the largest estimated $CO_2$ emissions, which are more than 60-fold higher than nuclear power.

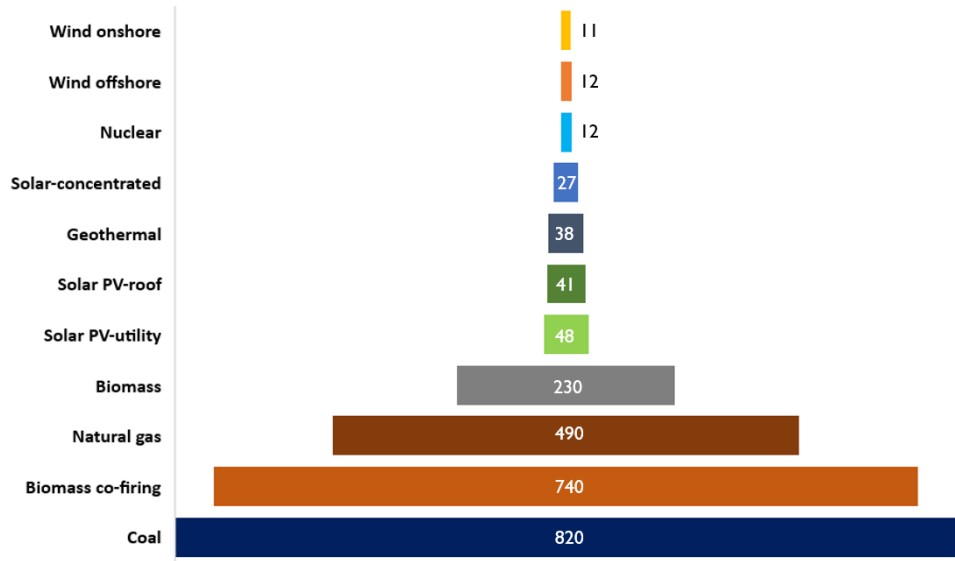

**Figure 1.** $CO_2$ emissions (g$CO_2$ equivalent/kW) of different electricity sources based on a life cycle analysis.

By the end of 2022, the United States (US) was reported as the largest supplier of global nuclear electricity generation; it generated 30% of the total nuclear energy generation, which is equivalent to 772 TWh, whereas China contributed half of that amount, at 16% of the total [4]. This capacity has also been contributed to by the initiation of nuclear reactors in Japan since the Fukushima Daiichi incident in 2011. However, the US's dependency on nuclear electricity was considered as minor compared to France, Russia, and South Korea. Nuclear electricity is the dominant source of power in France, where 63% of the electricity in the country was generated by nuclear sources, followed by Slovakia and Ukraine, at 59% and 58%, respectively [5]. The details of the nuclear electricity supplied by major countries in the world and the share of the total nuclear energy supply in the country are further illustrated in Figure 2.

The contribution of nuclear power to global electricity production is still considered as minor compared to the proportions from fossil fuels and renewable sources. It was recorded that the total worldwide nuclear electricity production by end of 2022 was 2611 TWh, which was only 9.2% of the total global electricity generation [5]. Additionally, no significant growth was observed between the years 2002 and 2022, where the annual growth average was about 0.2% [5]. The trend of global nuclear electricity production compared to fossil fuels and renewable sources in the past 5 years is shown in Figure 3. For instance, the global nuclear electricity generation in 2022 was observed to be lower by 4.7% compared to the previous year. This decline is mostly associated with the shutdown of many nuclear power plants for maintenance in France, Germany, and Japan, which exceed the planned shutdown schedule. The conflict in Ukraine has also forced several reactors to shut down,

which is associated with the loss of 21 TWh or 25% of the nuclear electricity produced in the country [5]. Nonetheless, South Korea, China, and Pakistan managed to increase nuclear electricity generation.

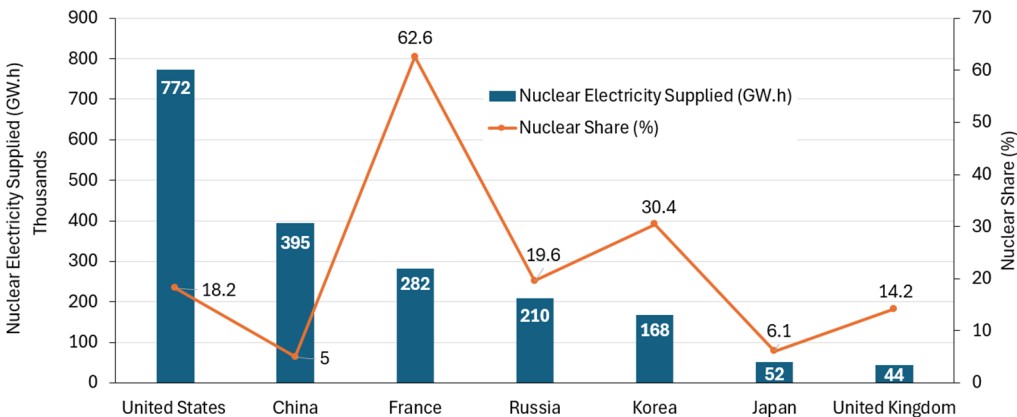

**Figure 2.** Supply of nuclear electricity and global nuclear share by major countries in the world as of 2022.

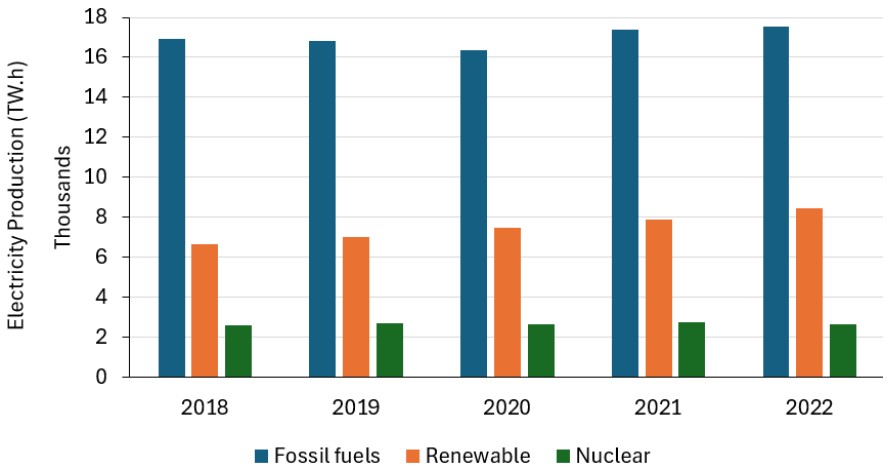

**Figure 3.** Comparison of electricity production from fossil fuels, renewable sources, and nuclear energy.

Nuclear generation is forecasted to increase by nearly 3% annually until 2026 due to ongoing maintenance completion in France, the resumption of nuclear production at multiple plants in Japan, and the commencement of commercial operations at new reactors across various markets such as in China, India, South Korea, and Europe [6]. Many nations are prioritizing nuclear power in their energy strategies to bolster energy security while curbing greenhouse gas emissions. Asia, particularly China and India, drives this growth, with the region's share of global nuclear generation expected to reach 30% by 2026. The commercial launch of China's first fourth-generation reactor in December 2023 further highlights the surge of China's global nuclear generation share from 5% in 2014 to around 16% in 2023 [6]. Overall, the world recorded a total of 437 operable nuclear reactors in 2022, which contributed to the total capacity of 394 GWe and more than 70% of the operable reactors used within 2018 until 2022 are of the pressurized water reactor (PWR) type [7]. Furthermore, six new reactors have been successfully connected to the grid, as recorded in China, Finland, Pakistan, South Korea, and the United Arab Emirates. Table 1 lists the operable nuclear power reactors that were available across the continents as of 2022.

**Table 1.** Number of operable nuclear power reactors by the end of 2022 [7].

| | Type of Nuclear Reactor | Asia | Western and Central Europe | North America | Eastern Europe and Russia | South America | Total |
|---|---|---|---|---|---|---|---|
| BWR | Boiling Light-Water Cooled and Moderated Reactor | 20 | 8 | 33 | | | 61 |
| FNR | Fast Neutron Reactor | | | | 2 | | 2 |
| GCR | Gas Cooled, Graphite Moderated Reactor | | 8 | | | | 8 |
| HTGR | High Temperature Gas Cooled Reactor | 1 | | | | | 1 |
| LWGR | Light-Water Cooled, Graphite Moderated Reactor | | | | 11 | | 11 |
| PHWR | Pressurized Heavy-Water Moderated and Cooled Reactor | 23 | 2 | 19 | | 3 | 47 |
| PWR | Pressurized Light-Water Moderated and Cooled Reactor | 104 | 98 | 61 | 40 | 2 | 307 |
| | Total | 148 | 116 | 113 | 53 | 5 | 437 * |

* Total is inclusive of 2 PWR units in Africa.

## 2. Introduction to Nuclear Fission and Fusion Reaction

Nuclear energy is produced from the core of an atom known as the nucleus, which contains protons and neutrons. Neutrons are surrounded by electrons that carry negative electrical charges and protons that carry positive electrical charges. The energy that holds the nucleus together is enormous and nuclear energy can be released by breaking those bonds, known as the fission reaction, splitting the nucleus into several parts [6].

The currently deployed nuclear power plants are employing nuclear fission reactions using uranium atoms [6]. The uranium atom (U-235) has an unstable particle arrangement and any excitement to the atom can disintegrate the nucleus. Figure 4 illustrates the splitting of U-235, which comprises 92 protons and 143 neutrons. Neutron bombardment into the nucleus of U-235 will split the atom and two or three neutrons are released each time the splitting happens, which creates the potential occurrence of chain reactions [7]. The atom splitting releases tremendous amounts of heat and radiation that can be harvested for energy generation.

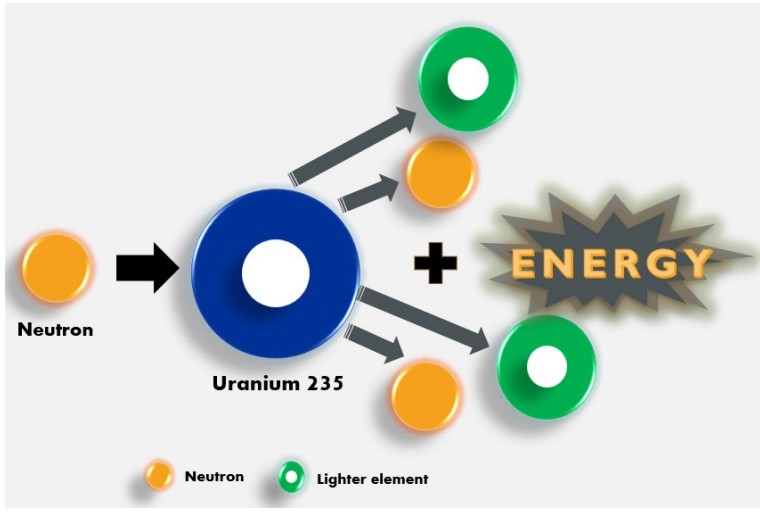

**Figure 4.** Splitting of uranium atom during the fission reaction.

The fission reaction poses a great risk for reactor meltdown due to the uncontrolled runaway chain reaction during the atom splitting. Therefore, comprehensive water treatment plants and radioactive waste management systems are crucial elements to the nuclear power plant for the safe disposal of radioactive wastes and all released materials. It is a national-scale disaster occurrence should an incident occur, which may lead to a nuclear explosion and the release of radiation. The history of nuclear incidents shows that radioactive contamination poses fatal threats to health and the surrounding areas, in both the short and long terms.

The known risks and challenges of performing fission reactions have promoted the transition to another means of producing nuclear energy, known as the fusion reaction. Figure 5 summarizes the overall advantages of fusion reactions. In contrast to fission reactions, fusion reactions take place by forcing two light nuclei together to form a heavier nucleus, which also releases an enormous amount of nuclear energy [8]. The fusion reaction is fundamentally safe, has no runaway chain reaction, utilizes fuels with shorter half-lives, and eliminates highly radioactive and long-life products [9]. The fusion reaction takes place in plasma, which is a hot gas containing positive ions and electrons that move freely [10].

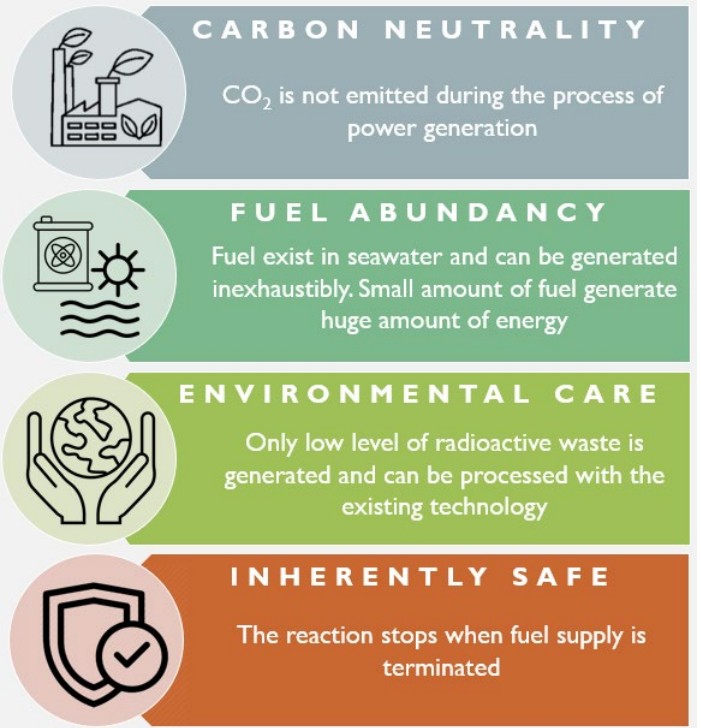

**Figure 5.** Advantageous properties offered by the nuclear fusion reaction.

Figure 6 demonstrates the fusion reaction using deuterium (D) and tritium (T), which releases the neutron, helium, and energy. Deuterium and tritium are hydrogen isotopes that serve as the common working gases in nuclear fusion reactions. The reaction yields a heavier helium nucleus and emits a neutron. Interestingly, the combined mass of the resulting helium nucleus and the neutron is slightly less than the combined mass of the original deuterium and tritium nuclei. Referring to Einstein's renowned formula, energy (E) is equal to the product of mass (m) and speed of light ($c^2$), the mass difference during the reaction is converted into a substantial amount of energy, which is introduced as the fusion energy [8]. Table 2 lists the fusion fuels that are most reported for the reaction and under study in predicting the reaction stability and energy generation [9]. It is estimated that 65% of the worldwide invested fusion programs are focusing on employing D-T fuels and a large majority of the programs (77%) are aiming for energy production [10].

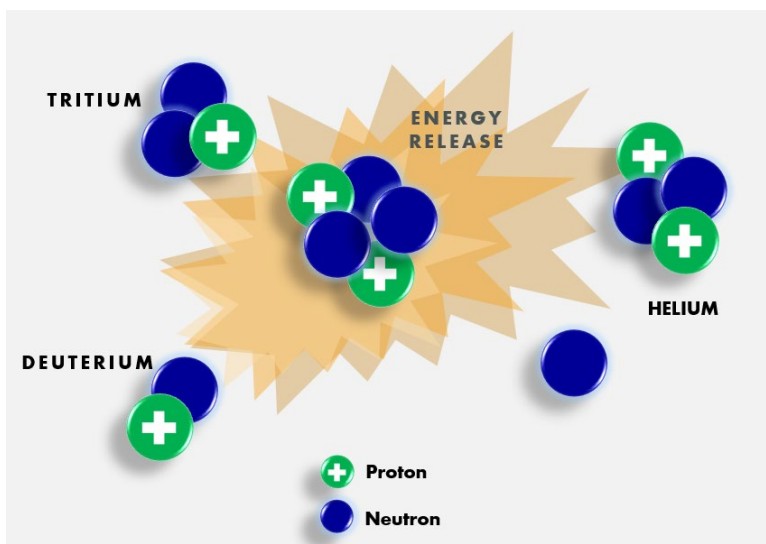

**Figure 6.** Fusion reaction using deuterium and tritium.

**Table 2.** Fusion fuels that are commonly applied for the fusion reaction demonstration.

| Reactants/Fuels | | Products |
|---|---|---|
| D | D | $^3$He (0.8 MeV) + n (2.45 MeV)<br>T (1.01 MeV) + p (3.02 MeV) |
| | T | $^4$He (3.5 MeV) + n (14.1 MeV) |
| | $^3$He | $^4$He (3.6 MeV) + p (14.7 MeV) |
| | $^6$Li | 2 $^4$He + 22.4 MeV |
| T | T | $^4$He + 2 n + 11.3 MeV |
| $^3$He | T | $^4$He + p + n + 12.1 MeV<br>$^4$He (4.8 MeV) + D (9.5 MeV) + p (11.9 MeV)<br>$^4$He (0.5 MeV) + n (1.9 MeV) + p (11.9 MeV) |
| | $^3$He | $^4$He + 2 p |
| | $^6$Li | 2 $^4$He + p + 16.9 MeV |
| p | $^6$Li | $^4$He (1.7 MeV) + $^3$He (2.3 MeV) |
| | $^{11}$B | 3 $^4$He + 8.7 MeV |

Fusion is considered as an energy-efficient technology. The amount of generated energy can be fourfold greater than from fission reactions [11] and approximately four million times higher than when burning coal or oil [12]. Theoretically, small amounts of fusion fuels (on a gram scale) can generate energy in terajoule scales, which is the sum of the energy per capita for sixty years in a developed country [9]. It is estimated that the energy yield from a kilogram of fusion fuel is equivalent to 10 million kilograms of fossil fuel [13]. Research on controlled nuclear fusion and plasma physics is currently conducted in over 50 member states of the International Atomic Energy Agency (IAEA). The objective is to demonstrate the scientific viability of fusion as a potential energy source [14].

Joint European Torus (JET) has successfully demonstrated its D-T fusion experiment; it was able to produce 59 MJ of energy over a five-second duration by burning only 170 micrograms of the deuterium–tritium fuel and yielded an average fusion power of 11 MW [13]. In December 2022, the Lawrence Livermore National Laboratory (LLNL) successfully obtained a positive energy return on investment (EROI). The conducted experiment managed to surpass the threshold energy by achieving 3.15 MJ of fusion energy output by delivering 2.05 MJ of energy to the target [15]. It is a ground-breaking finding that further shifted the expectation of fusion as a viable source of power and electricity generation by 2055.

This goal is indicated in the EU fusion roadmap and this achievement has ramped up the momentum of nuclear fusion toward rapid progress for commercialization [16].

Nuclear fusion was first introduced into international dialogue during the 26th Conference of Parties (COP26) in 2021 by the United Nations Framework Convention on Climate Change (UNFCC) [17] (this was almost 70 years after it was first initiated in 1950) as having the potential to curtail climate change and be used as a nuclear weapon [18]. Fusion technology has diverse market sectors and can significantly impact the national and global economy. For instance, electricity generation is the most interesting market sector aimed by the private sector, whereas industrial heat and hydrogen/clean fuels are considered the most potential spin-off markets to invest in [19]. However, there are other market sectors with the potential for booming, such as off-grid energy, medical uses, as well as space and marine propulsion. The findings of a market survey of the private sector on the market potential of fusion technology are illustrated in Figure 7.

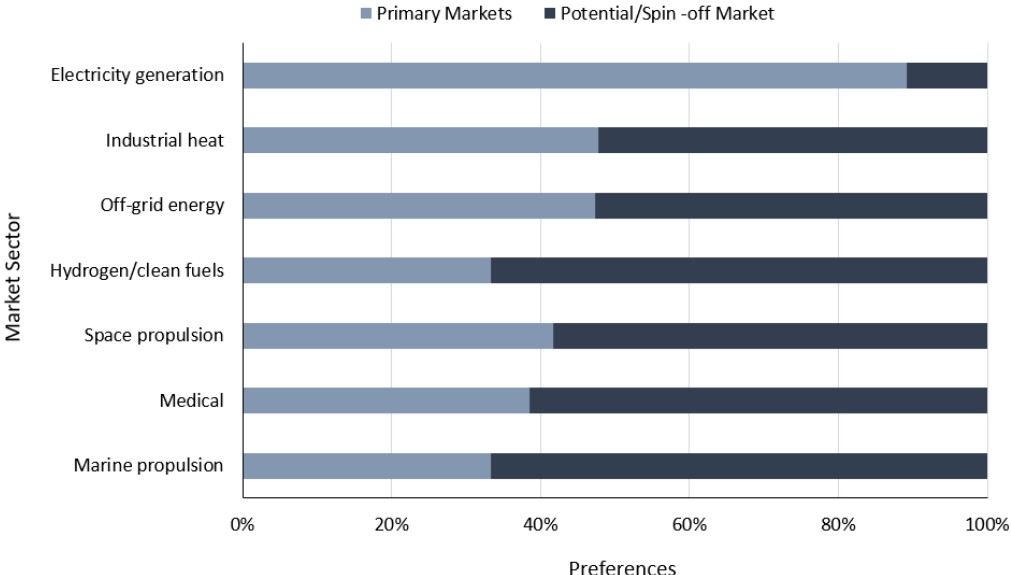

**Figure 7.** Preferences of market sector aims by private sectors for fusion technology, as surveyed in 2023.

## 3. Development of Fusion Plants: Scale and Readiness

### 3.1. Overview of the Fusion Device

In 2022, the International Atomic Energy Agency (IAEA) recorded a total of 143 fusion devices around the world that are operational, under construction, and planned; this total comprises 131 experimental designs and 12 DEMO designs. The configuration of the designs is defined by four categories, tokamaks, stellarators/heliotrons, laser/initial, and alternative concepts such as dense plasma focus, inertial electrostatic fusion, magnetized target fusion, reverse field pinch, and spheromak [20]. To date, all operating fusion devices are experimental designs, a total of 98, whereas 11 experimental designs are under construction. Conversely, all DEMO fusion devices are still only planned and not built. In 2023, there were nine tokamak configuration devices that were planned to be established at a DEMO scale, with six of them affiliated to public ownership. Figure 8 shows the classification of worldwide fusion devices according to the design and configuration, whereas Figure 9 describes the status of the designed fusion devices. These outlooks suggest that fusion devices that are globally ready for the current scenario exist at an experimental scale, and tokamak is depicted as the most established experimental design configuration, while global interest in alternative designs is growing.

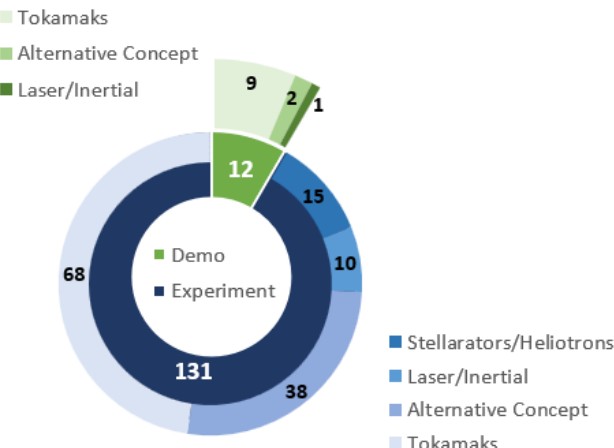

**Figure 8.** Number of fusion devices (unit) according to scale and design configuration worldwide.

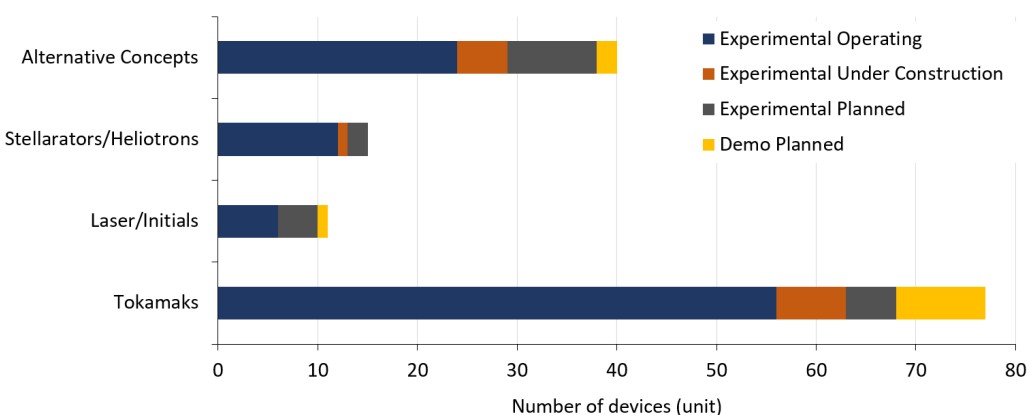

**Figure 9.** Status of fusion devices by design configuration.

Tokamak is a Russian acronym meaning "toroidal chamber with magnetic coils" [21]. It is a magnetic confinement fusion (MCF) device that offers superior confinement capabilities and simple construction. MCF applies a magnetic field as the containment mechanism to hold hot plasma at temperatures exceeding 10 keV in the device [8]. The magnetic mirror represents the most basic design for these devices, yet it exhibits limited confinement capability. Intense magnetic fields in tokamaks are generated via three magnetic field configurations by the coils at the torus, which are referred to as toroidal, poloidal, and helical magnetic fields [22]. The toroidal magnetic field is directed in the longitude direction around the torus, whereas the poloidal magnetic field is directed in the short direction around the torus. These two magnetic fields result in twisted magnetic fields that confine the plasma [23]. The shape and position of the plasma are controlled by the outer poloidal coil via generation of the outer poloidal field. The helical magnetic field is formed in the tokamak due to the toroidal coils that generate the toroidal field and also due to the plasma current that produces the poloidal field [24]. Conversely, the stellarator configuration device offers a superior confinement performance that facilitates steady-state operation with fewer magnetohydrodynamic (MHD) instabilities and minimal disruption occurrences [8,25]. However, the stellarator has a highly intricate design that poses significant challenges for manufacturing.

The world's largest operating tokamak experiment has been established through the Joint European Torus (JET), and the device can hold a mega-ampere of plasma currents. JET is also referred to as the experimental flagship of fusion technology in Europe [13]. However, the highest number of available tokamaks in the world in 2023 was in Japan, which had a total unit of 13 devices, followed by China, US, Russia, and the UK [20]. The largest proportion of the total number of fusion device units with all types of design

configurations is located in the US which has 30.4% of the total, equivalent to 34 devices, half of which are fusion devices of various alternative concept configurations [20]. The list of operational tokamak experiments around the world is summarized in Table 3, whereas Figure 10 compares the number of fusion devices according to design configuration in the top 10 countries of fusion technology.

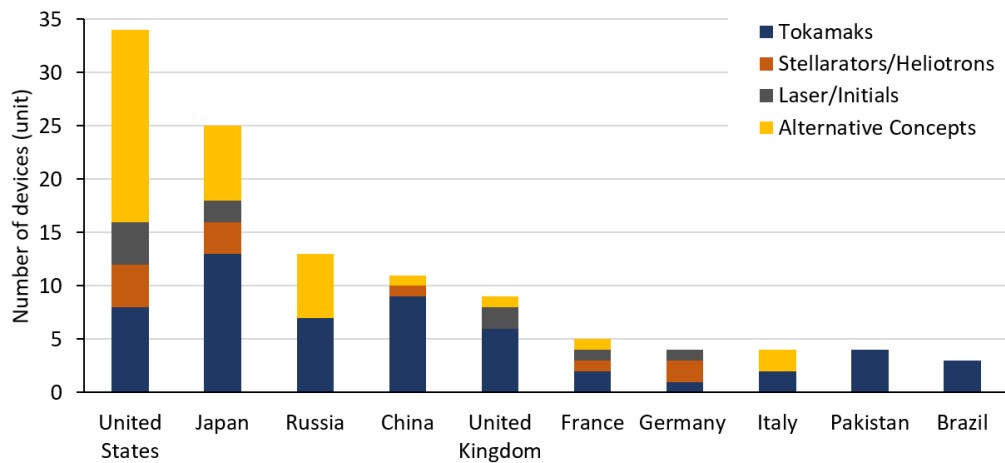

**Figure 10.** Top 10 countries with planned, under construction, and operating fusion devices.

**Table 3.** Operating Experimental Tokamaks as of 2022 [9].

| Country | Type | Number of Devices |
|---|---|---|
| Brazil | Conventional Tokamak | 2 |
| | Spherical Tokamak | 1 |
| Canada | Conventional Tokamak | 1 |
| China | Conventional Tokamak | 4 |
| | Spherical Tokamak | 2 |
| Costa Rica | Spherical Tokamak | 1 |
| Czech Republic | Conventional Tokamak | 1 |
| Denmark | Spherical Tokamak | 1 |
| Egypt | Conventional Tokamak | 1 |
| France | Conventional Tokamak | 1 |
| Germany | Conventional Tokamak | 1 |
| India | Conventional Tokamak | 2 |
| Iran | Conventional Tokamak | 3 |
| Italy | Conventional Tokamak | 1 |
| Japan | Conventional Tokamak | 5 |
| | Spherical Tokamak | 4 |
| Kazakhstan | Spherical Tokamak | 1 |
| Libya | Conventional Tokamak | 1 |
| Pakistan | Spherical Tokamak | 2 |
| Portugal | Conventional Tokamak | 1 |
| South Korea | Conventional Tokamak | 1 |
| | Spherical Tokamak | 1 |

**Table 3.** *Cont.*

| Country | Type | Number of Devices |
|---|---|---|
| Russia | Conventional Tokamak | 3 |
| | Spherical Tokamak | 2 |
| Switzerland | Conventional Tokamak | 1 |
| United Kingdom | Conventional Tokamak | 1 |
| | Spherical Tokamak | 2 |
| United States | Conventional Tokamak | 2 |
| | Spherical Tokamak | 3 |

*3.2. Establishment of the International Thermonuclear Experimental Reactor (ITER) and DEMO*

The great international joint experimental works known as the International Thermonuclear Experimental Reactor or ITER, was inaugurated due to the challenges observed in JET [13]. The challenges arise due to the high auxiliary power requirements for heating the systems and energizing the magnetic coils, limitations that led to the usage of superconducting magnets in ITER. ITER is being built in Southern France through a collaboration among 35 nations including those in the European Union, China, India, Japan, South Korea, Russia, and US [26]. Its primary objective is to investigate and demonstrate plasma burning that can minimize or eliminate the need for the external heating of fusion devices [26]. The scientific goal of ITER is to prove the viability of generating 500 MW of fusion power with a Q factor of 10 and sustaining it for 400 s [27]. The tokamak at ITER is expected to produce 13 tesla of superconducting magnet field, equivalent to 280 thousand times the Earth's magnetic field [23]. The construction of the ITER facility is based on detailed safety studies and considerations, which were evaluated and authorized by the French Nuclear Safety Authority [28]. It is structured to investigate the capability for tritium breeding, since the tritium self-sufficiency and its capacity are still obscure.

The assembly and integration work for ITER commenced in July 2020 and, to date, the program has faced multiple challenges to achieve its objectives. Among the crucial challenges included managing off-normal events, comprehending the confinement and transport physics of the burning plasma, addressing the high-power particle and heat flux exhaust issues on the diverter, understanding energetic particle behavior, achieving high-performance long pulse and steady-state operation, and ensuring successful tritium breeding and retention [27]. However, the physical completion percentage for the first plasma continues to grow and had managed to achieve 77.0% progress by the end of June 2022 [29]. The European nations shoulder most of the construction costs (45.6%), with the remaining expenses evenly distributed among China, India, Japan, South Korea, Russia, and the US (9.1% each) [30].

ITER's program is based on a scientific and technological research facility focusing on a tokamak with a single-null poloidal divertor [31]. Its toroidal field (TF) coils adopt a D-shaped configuration, whereas the reactor operates with vertically elongated plasmas. The tokamak structure measures 29 m in diameter and 30 m in height. The following are among the milestones charted through ITER [31]:

- Attain a quasi-stationary plasma discharge that is sustained under inductive discharge conditions.
- Validate the achievement of stationary operation through non-inductive discharge current drive.
- Demonstrate the technical feasibility and conduct training exercises by utilizing fusion technologies and equipment.
- Evaluate key functional components intended for future reactors.
- Assess materials and equipment within a fusion-neutron-irradiation environment.
- Validate blanket concepts and investigate various tritium breeding modules.

The success of the heating power generated by ITER will be translated into electricity for the grid by 2050, known as DEMO [13]. ITER will bridge the gap for DEMO establishment and subsequently be geared towards attaining the tritium self-sustainability of the planned DEMO [32]. It will pave the way for the future of fusion power in terms of the commercialization and production of a cost-efficient and carbon-free energy market [33]. Figure 11 visualizes the milestone schedule of both ITER and DEMO, as well as the integration of information from activities conducted at ITER towards the establishment of DEMO [16]. DEMO initiated its pre-conceptual design in 2014 and is expected to achieve its final milestone by 2060, when the commissioning and operation of DEMO take place. In the current year, DEMO is at the second milestone, the conceptual design, which is scheduled to be completed by 2027. Meanwhile, ITER is expected to initiate its operation in 2025, using the first plasma, and be able to generate data for DEMO from six major activities within 15 years of its program [16]. DEMO will be operational around 20 years after high-power burning plasmas are demonstrated in ITER; ITER should be functional for high-power operation by the end of 2040 [16]. Overall, the roadmap is set to achieve its ultimate goal by the second half of the century.

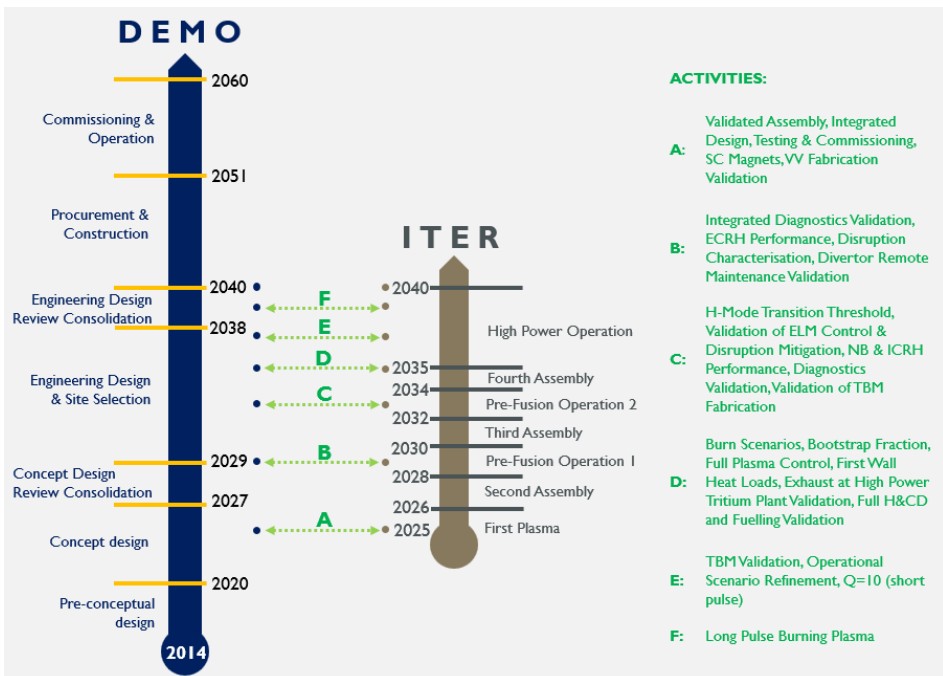

**Figure 11.** Milestone schedules of DEMO and ITER and integration of data from activities by ITER into DEMO.

DEMO is expected to include tritium breeding for fusion fuel, demonstrate suitable material for neutron handling, and illustrate the requirement of safety, environmental sustainability, and sufficient technology for an established commercial power plant. For instance, the European union aims to initiate the major engineering design for DEMO by 2030 and have the plant ready for operation by 2040 [16]. The plan for DEMO establishment in other countries using tokamak configurations and their design properties are summarized in Tables 4 and 5, respectively.

**Table 4.** Plan for DEMO fusion devices using tokamak configurations by various countries [20].

| Country | Organization | Device Name | Details of Construction |
|---|---|---|---|
| United Kingdom | UKAEA | STEP | The first phase is producing a design concept by March 2024 and the construction of the prototype power plant aims to be completed by 2040. |
| | Tokamak Energy | DST-E1 | |
| United States | Commonwealth Fusion System | ARC | Aims to bring fusion for commercial viability during 2035–2040 by positioning the country as a leader in fusion and speeding up the transition to low-carbon energy by 2050. |
| | General Atomics | GA-FPP | |
| China | Chinese Consortium | CFETR | Bridge the gap between ITER and DEMOs, where the construction started in the 2020s and there will be construction of a DEMO in the 2030s. |
| European Union | Eurofusion | EU-DEMO | Demonstrate technology and economic viability of fusion through hundreds of MWs of net electricity. It is currently in the conceptual design phase (2021–2027). |
| Japan | Japanese Consortium | JA-DEMO | Construction is planned to start around 2035. |
| South Korea | Korea Institute of Fusion Energy | K-DEMO | The conceptual design study was initiated in 2012 and aims for construction by 2037. The first phase (2037–2050) device will be used for component development and testing and will be applied for phase two, where the demonstration of net electricity generation is by 2050. |
| Russia | Russian Consortium | STEP | It aims to harvest fusion-produced neutrons to turn uranium into nuclear fuel and destroy radioactive waste. It is planned to be built by 2023 and is part of the country's fast-track strategy to establish a fusion power plant by 2050. |

**Table 5.** Design properties of DEMO tokamaks from China, South Korea, Japan, the US, and the EU as compared to ITER [9].

| Device Name | Radius of Plasma (R, m) | | Central Magnetic Field ($B_t$, T) | Elongation Ratio ($\kappa$) | Plasma Current ($I_p$, MA) | Fusion Power (GW) |
|---|---|---|---|---|---|---|
| | Major | Minor | | | | |
| ITER | 6.2 | 2.0 | 5.3 | 1.7 | 15 | 0.5 |
| EU-DEMO | 9.1 | 2.9 | 5.86 | 1.65 | 17.75 | 2.0 |
| CFETR | 7.2 | 2.2 | 6.5 | 2.0 | 13 | 1 |
| K-DEMO | 6.8 | 2.1 | 7.4 | 1.8 | >12 | ≈3 |
| JA-DEMO | 8.5 | 2.42 | 5.94 | 1.65 | 12.3 | 1.46 |
| ARC | 3.3 | 1.1 | 9.2 | 1.84 | 7.8 | 0.53 |

### 3.3. Safety Standards for Nuclear Fusion Energy Technology Development

As the power generation through fusion energy emerges as a promising, clean, and sustainable future energy resource, its practical implementation is potentially associated with radiological and industrial hazards. These concerning risks necessitate the development of safety standards to address such hazards, ensuring the safety of both people and the environment, as well as to foster public trust in nuclear power generation programs. The inadequacy of legislative and regulatory frameworks for engineering design may lead to major safety accidents that can cause detrimental impacts and fatalities [34].

The American Society of Mechanical Engineers (ASME) Boiler and Pressure Vessel (BPVC) and the French Association for Design, Construction and In-Service Inspection Rules for Nuclear Steam Supply Components (AFCEN) RCC/RCC-MRx are among the common industrial standards widely used as the basis for the engineering designs of mechanical components in fusion reactors [35]. In 2023, ASME BPVC Section III Division 4 was

issued to provide requirements for the construction of fusion energy devices, specifically for commercial applications on a global scale. The standard covers two primary fusion device concepts: magnetic confinement fusion (MCF), like tokamaks, and inertial confinement fusion (ICF) [3]. This code and standard will serve as a universal basis for the construction, licensing, and operation of fusion facilities, such as compact pilot plants and DEMOs. Within Section III, this code specifically provides the guidelines on the design, manufacturing, and construction of fusion-related components such as vacuum vessels, cryostats, and superconductor structures and their interactions with each other. Additionally, the standard also covers on the support structures, materials, containment structures, piping, vessels, valves, pumps, and supports related to fusion energy devices. The development of these codes and standards will provide a strong foundation for the advancement of the fusion industry.

On the other hand, in China, the Nuclear Safety Law 2017 and the Law on Prevention and Control of Radioactive Pollution 2003 established the foundation for nuclear safety to safeguard the engineering design of CEFTR [36]. Moreover, the safety regulations for nuclear energy in China have also been complemented by seven administrative regulations approved by the State Council. Additionally, national standards and industry standards, such as GB18871-2002 (basic standards for protection against ionizing radiation and the safety of radiation sources) and GB6249-2011 (standards for the environmental radiation protection of a nuclear power plant), provide guidelines for radiation protection in nuclear facilities [36]. Meanwhile, for ITER, the licensing process involved approval by the French Nuclear Safety Authority (ASN) as a basic nuclear installation (INB) after the submission of project, environment, and safety reports [37,38]. The involvement of the ASN in the licensing process for ITER includes reviewing safety files submitted by the ITER organization such as the preliminary safety report (Rapport Préliminaire de Sûreté, RPrS). RPrS covers an extensive safety analyses and assessments that are scrutinized by the ASN and its technical advisors These regulatory frameworks and licensing procedures are essential for ensuring nuclear safety and compliance with standards in both China and ITER.

### *3.4. The United Kingdom (UK)*

Fusion in Europe was initiated in 1957 by the Euratom Treaty, which recognized the European Atomic Energy Committee to ensure the acceleration of fusion technology in the region [39]. The collaboration of major fusion laboratories in Europe happens through a consortium known as EUROfusion, and the design of JET was initiated through European collaboration by making Culham in Oxfordshire the host of the project [17]. Since then, JET has been the largest operational magnetic confinement plasma physics experiment in the world; it was completed in 1983 and achieved its first plasma [18]. Furthermore, the Culham Centre for Fusion Energy (CCFE) in Oxfordshire has been the base of the UK fusion research program and the fusion research arm of the UK Atomic Energy Authority (UKAEA). Most of the research works at the CCFE are funded under the Euratom Treaty through the Engineering and Physical Sciences Research Council and European Union [13]. The UK Department for Energy Security and Net Zero assigned and sponsored the UKAEA as the national body that is responsible for the research and execution of fusion technology in the country [40]. For instance, GBP 650 million has been utilized by the UKAEA for the Fusion Futures Program to support the UK Fusion Strategy and establish new facilities at the CCFE that will accelerate the advancements in fusion fuel cycle technology [40]. The UK Fusion Strategy was planned for timeframe from 2022 to 2026, with the mission of leading sustainable fusion energy delivery and maximizing the benefits scientifically and economically [41]. Four strategic goals have been incorporated into the strategy, which are illustrated in Figure 12.

EUROfusion has rolled out the fusion energy roadmap in Europe to build fusion power plants and produce electricity for commercialization through the establishment of JET in the UK [16]. Figure 13 illustrates the strategies for the fusion research roadmap in Europe to establish a fusion power plant [13]. However, the operational phase of the JET fusion

experiment was terminated by the end of 2023 [42]. The decommissioning of JET presents a distinctive chance for the UK to gain insights into the challenges associated with this process. It will strengthen the country's plans for future commercial fusion facilities and help to develop intellectual property and maintain the necessary skills for decommissioning future fusion power plants [42]. The UK has launched the JET decommissioning and repurposing (JDR) program to elucidate the entire life cycle of a fusion power plant [43]. It will foster groundbreaking research and innovation, defining the prerequisites for the safe and ethical execution of fusion decommissioning and repurposing efforts. As of 2023, the JDR program is progressing toward submitting an outline business case (OBC) to the Department for Business, Energy, and Industrial Strategy (BEIS) [43].

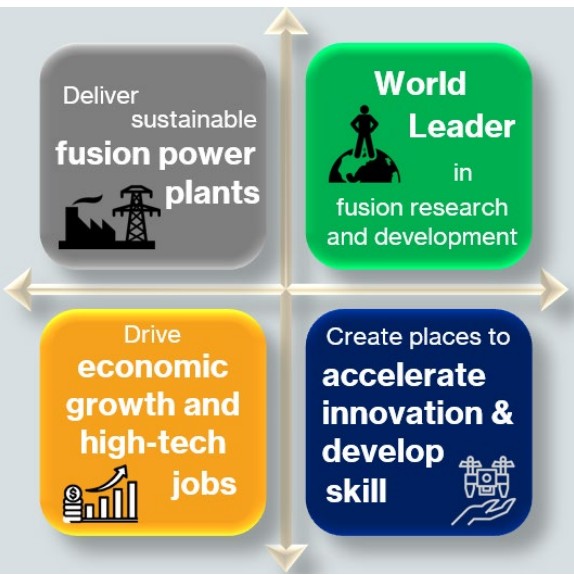

**Figure 12.** Strategic goals of the UK Fusion Strategy.

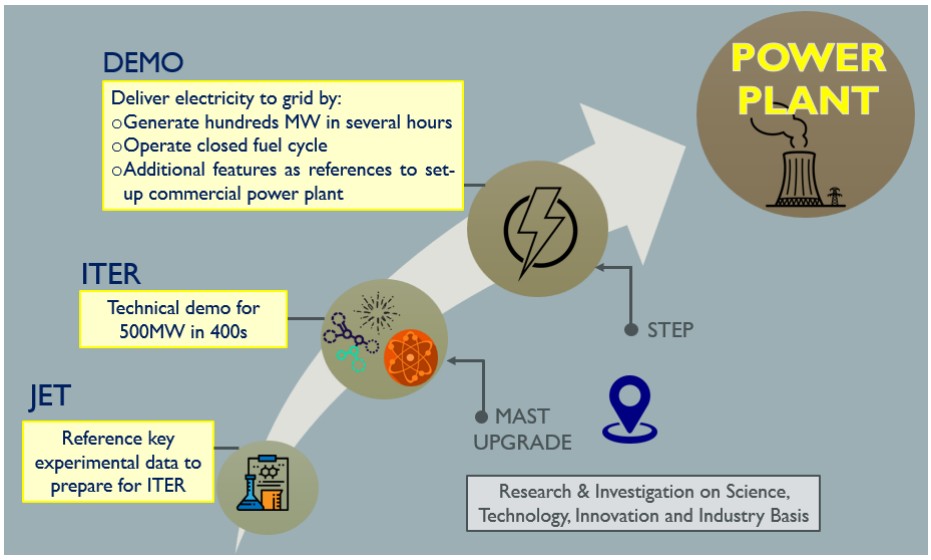

**Figure 13.** Summary of the roadmap strategy by EUROfusion to achieve a fusion power plant by the middle of the century.

The UK government announced the founding of UK Industrial Fusion Solutions Ltd. to build a prototype fusion power plant known as spherical tokamak for energy production (STEP) by 2040 [10].The announcement also came with specific regulations on fusion that exclude nuclear licensing requirements. The government has already allocated over GBP

240 m for the initial phase of STEP up to 2024, with an additional commitment bringing the total funding to over GBP 300 m by 2025 [42]. The forthcoming tranche, starting in 2024, will yield independent benefits such as fostering skills and expertise, generating data and intellectual property, and bolstering UK supply chains [42]. STEP's pivotal role lies in showcasing the commercial feasibility of fusion through the operation of large-scale fusion systems within a single energy-producing facility, thus catalyzing the entire fusion sector in the UK. It will demonstrate the ability of fusion to generate net electricity, allow familiarization with the operational handling of fusion power plants, and authenticate the production of fusion fuel [44]. As described in Figure 14, STEP will be conducted in three phases, where the tokamaks will be connected to the National Grid, although it is yet to be at the commercial stage [45]. As of now, STEP is concluding its Phase I, which is to produce a concept design of a compact "spherical tokamak" reactor concept for a fusion power plant by 2024 and, thus, STEP can be considered as one of the reference fusion facilities for the concept design of an integrated plant sometime this year.



**Figure 14.** Phases of development program for STEP.

The UKAEA's Mega-Amp Spherical Tokamak-Upgrade (MAST-U) represents an innovative approach to fusion research and aimed at exploring the feasibility of fusion power production in a more cost-effective and scalable manner [42]. The original MAST experiment was in operation from 2000 to 2013 and demonstrated a reliable performance that constituted a significant upgrade once completed in 2020. Its enhanced version, called MAST Upgrade, has major new capabilities such as plasma capabilities and an exhaust system [16]. This upgraded version was finalized in 2021 and has yielded a tenfold reduction in the heat transferred from the plasma to the surrounding components [42]. MAST-U and STEP will provide significant support to the implementation of ITER and DEMO. In the next plan, the UK government aims to solve the challenges of fusion commercialization by building a fusion fuel cycle facility via a partnership with academics and industry. The government also plans to utilize the lesson learnt by JDR to develop technical skills and expertise for a future fusion power plan as well as finding new alternatives for collaboration and expert sharing with ITER [42].

### 3.5. The United States (US)

The United States (US) has set the goal of achieving net-zero emissions by 2050, with a dual focus on bolstering energy security and fortifying America's technological competitive advantage. For decades, the country has invested in its domestic fusion program, primarily spearheaded by the Department of Energy National Laboratories, and actively participated in significant international collaborations in the field of fusion research. For instance, the US National Ignition Facility (NIF), through the Lawrence Livermore National Laboratory (LLNL), engaged in firing laser beams onto frozen pallets of deuterium and tritium in diamond capsules that were suspended inside a gold cylinder [46]. The laboratory managed to perform the first experiment that reached "ignition" in December 2022, where 54% more energy was released than energy consumed, and at least three more ignitions have been obtained since then [46]. Based on this breakthrough in the physics of magnetically confined plasmas, the US aims for fusion pilot plants by the 2040s, with the vision of economically attractive fusion energy [47]. In 2023, the US had the highest number of fusion devices in the world. The US has a total of 34 fusion devices (including those

planned, under construction, and currently operating), where 5 devices will be developed as DEMO. The largest device configuration interest in US is for alternative concepts, followed by tokamaks [20]. Figure 15 summarizes all the fusion devices in the US according to the device configuration, status of construction, and device ownership.

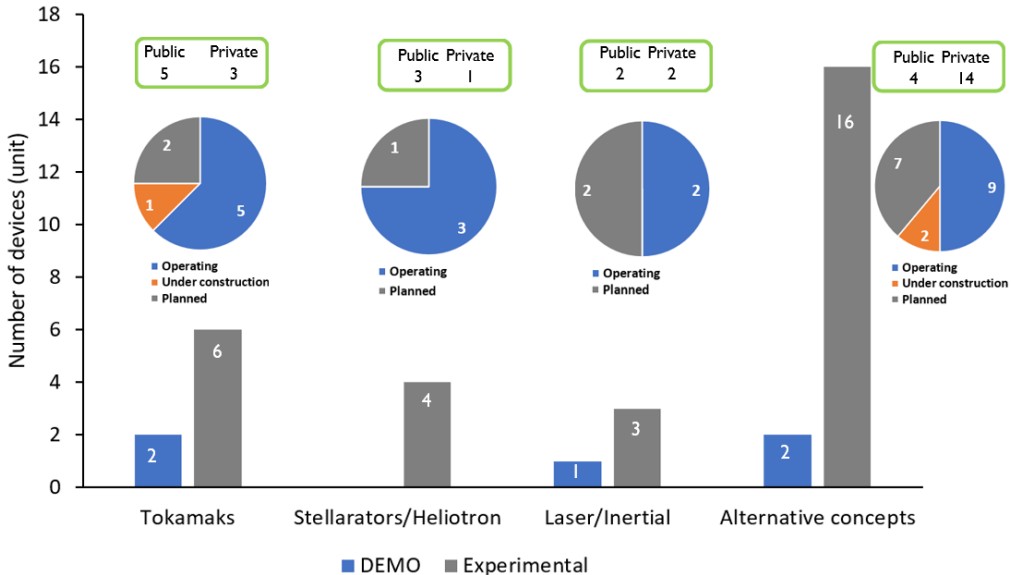

**Figure 15.** Number and status of fusion devices in the US.

The US Department of Energy (DOE) established the Fusion Energy Sciences (FES) program under the Office of Science, with the mission of gathering scientific and engineering expertise. Four research elements were introduced to support the mission, based on the concept of plasma sciences [48]. Each research element incorporates the activities of targeted scientific research regimes that can contribute towards the implementation of the overall US fusion program accordingly. The details of each research element in FES are described in Figure 16. US-DOE also co-hosted its first summit on fusion in early 2022, where the government announced three new initiatives towards developing commercial fusion energy as follows [49]:

- Community engagement: establish a decadal strategy to promote the understanding of commercial fusion energy by all stakeholders.
- Launch of a wide fusion initiative: aim to accelerate the feasibility of commercial fusion energy in the private sector.
- Funding opportunities: two funding amounts of USD 50 million to support advanced technology and scientific research on the fusion pilot plant.

Private sector involvement in fusion R&D has been observed to have increased dramatically in the US since, to mitigate the challenges in developing fusion energy, it was opened up for investment in 2019. In 2020, the US Congress increased the financial allocation for the fusion programs to be performed at Inertial Confinement Fusion Mission National Laboratories [34]. Currently, the largest new fusion research work in the US is being conducted by the private sector, where both the 3- and 5-year moving averages investment in the country have exceeded the annual budget allocated for the FES program [50]. The US has become the home for 43% of the headquarters of global private companies that invest in fusion technology [19].

Multiple kinds of fusion device configurations are planned, developed, and operated in various fusion facilities. One of the DEMO tokamak fusion devices planned for operation in the US, known as the affordable, robust, compact (ARC) reactor has been developed at Massachusetts Institute of Technology (MIT). It is designed to be functionally flexible for use in both fusion power plants and fusion nuclear science facility (FNSF) testing in D-T environments [51]. The ARC reactor offers an overall cost reduction in reactor building

due to its smaller size and allows for modification in the device operation and functionality due to its modular nature. The flexible properties of ARC enable innovative operational design and minimize the cost as well as the risk of testing failures. It also employs fluorine lithium beryllium (FLiBe) molten salt, which permits an output blanket temperature of 900 K and has high efficiency in the helium Brayton cycle, which permits the generation of net electricity while operating as a pilot power plant [51].

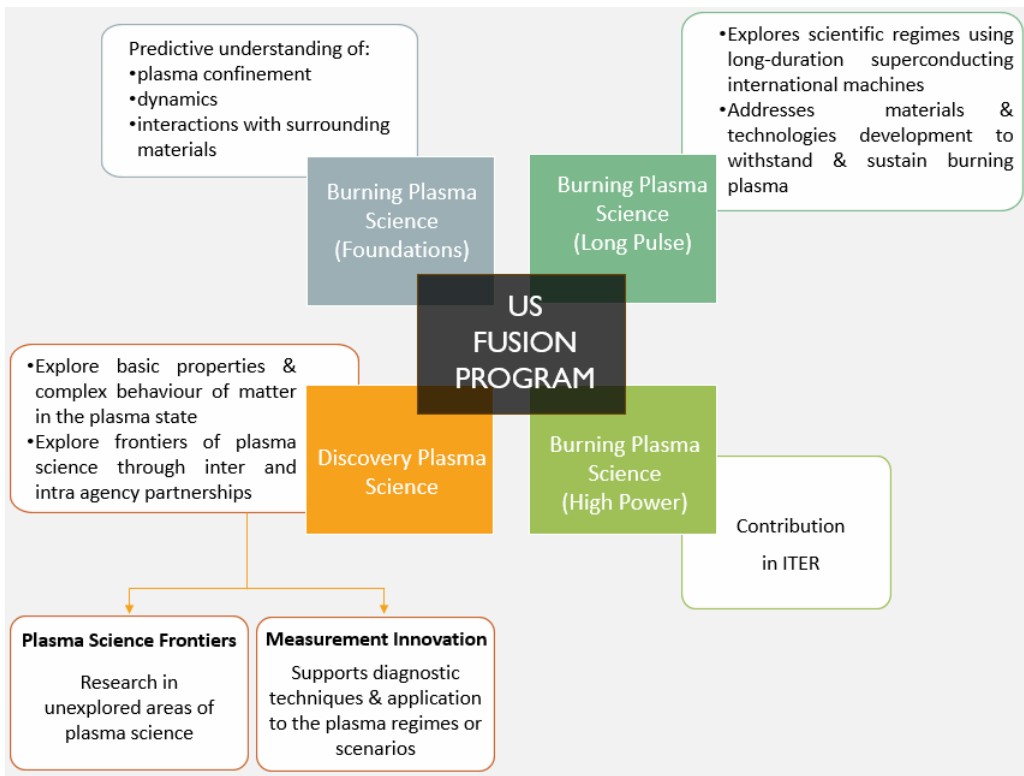

**Figure 16.** The elements in the FES of the US Fusion Program.

The technology and physics applied by ARC has been further verified by the SPARC tokamak project. SPARC demonstrates burning-plasma regimes in high fields for fusion power plants and exploits the enhancement of high-temperature superconductor (HTS) materials that lead to the advancement of fusion technology and facility construction. As such, SPARC aims to demonstrate the viability of rare earth barium copper oxide (REBCO) HTS magnet application in an integrated fusion confinement facility [52]. REBCO offers high magnetic fields that allow the device to be built at a smaller size, thus accelerating the completion of device construction and enhancing the cost effectiveness of the reactor [53]. SPARC is also targeted to achieve a fusion gain (Q) larger than 2, which is the critical indicator to pave the way for commercializing fusion energy [52]. The construction of the device was initiated the middle of 2021, and it is expected to generate a fusion power of 140 MW with plasma gain of $Q \approx 11$ [52].

The US has also intensified its fusion program through strategic partnerships with the European region. The country sealed a strategic partnership with the Department for Energy Security and Net Zero of the United Kingdom and Northern Ireland (DESNZ) to strengthen the US's bold decadal vision for commercial fusion energy and, additionally, the UK's Fusion Strategy. This partnership has accelerated the demonstration and commercialization of fusion energy in both countries [54]. For instance, the UK and US partnerships have involved UKAEA–Princeton Plasma Physics Laboratory Fusion Fellowships, MAST-U tokamak, and the DIII-D National Fusion Facility [54].

The US is an active member of ITER by supplying 9.1% of the overall ITER construction cost [30]. The ITER project is succeeded by Oak Ridge National Laboratory, with two partner

laboratories known as the Princeton Plasma Physics Laboratory and the Savannah River National Laboratory, where their mission is to design, construct, and assemble a burning plasma experiment [55]. The country is fully responsible for contributing the hardware for ITER, such as vacuum auxiliary and roughing pump systems, central solenoids, pellet inject (fueling) systems, tokamak exhaust processing systems, disruption mitigation systems, and tokamak cooling water systems [55]. To date, the US has managed to accomplish 75% of its project milestones by delivering the essential parts for energizing the pumps and auxiliary systems at the ITER facility, known as the steady-state electrical network (SSEN). The country also concluded its final review of the design of the tokamak cooling water system for ITER in October 2023. Thus, the US is actively progressing towards the establishment and operation of ITER.

*3.6. China*

China is the fourth country that has develop superconducting tokamaks through its advanced research on HT-7 tokamaks, which led to the foundation of research into magnetic confinement fusion (MCF) and the roll out of a fusion energy roadmap [25]. The country strengthens its commitment to fusion, as influenced by the breakthrough achievement of the NLC in the US, which attained net fusion energy and the announcement of the fusion roadmap by the EU to realize fusion plants by 2055. It was reported that a research facility in China implemented a high-confinement operation mode with a plasma current in August 2023, which is considered the first implementation in the world [56]. Currently, six tokamaks are operable in China at the experimental scale, the details of which are summarized in Table 6. Most fusion devices in China are of the tokamak configuration, where one tokamak is planned for DEMO by the Chinese Consortium, known as the China Fusion Engineering Test Reactor (CFETR).

**Table 6.** The available tokamaks currently operating in China at an experimental scale [20].

| Tokamak Type | Device Name | Organization |
|---|---|---|
| Conventional | Experimental Advanced Superconducting Tokamak (EAST) | Institute of Plasma Physics, Chinese Academy of Sciences |
| | Huan-Liuqi-2A (HL-2A) | Southwestern Institute of Physics |
| | Huan-Liu-3 (HL-3) | |
| | Joint Texas Experimental Tokamak (J-TEXT) | Huazhong University of Science and Technology |
| Spherical | Sino-United Spherical Tokamak (SUNIST-1) | Tsinghua University |
| | XuanLong Experiment (EXL-50) | ENN |

CFETR is a magnetic confinement fusion device that is planned to realize commercial fusion energy in China and bridge the gaps between ITER and EU-DEMO [57]. The conceptual design of this conventional tokamak was completed in 2015, and its construction is expected to be accomplished by 2040 [9]. CFETR aims to obtain a net engineering gain of higher than 1 ($Q_{eng} > 1$) [9]. The R&D of CFETR will be executed in two phases, where the first phase will focus on achieving steady-state operation and self-sufficiency with fusion power up to 200 MW, whereas the second phase will focus on validating relevant issues for DEMO operation with fusion power above 1 GW [9].

One of the challenges of executing CFETR is the graded conductor design of the toroidal field (TF) magnet, which can be resolved by having a TF magnet with a high current carrying capacity and strong magnetic field [32]. The higher magnetic field is favorable for the tokamak due to better confinement and fewer transportation issues, despite difficulties in its engineering design. In addition, low fusion power is required for steady-state operation and tritium breeding, which eliminates the necessity of having

high-Q-burning plasma [27]. An efficient auxiliary drive and high bootstrap current can generate the reliable steady-state operation of a DEMO [27]. Wan et al. [58] estimated more fusion power and a better steady-state operation was demonstrated by relatively low density with a normalized energy confinement time $H_{98}$ of less than 1.3, due to a higher current drive efficiency and large effective collision. The tritium self-sufficiency device is also achievable by having a high fuel-burning rate, advanced tritium breeding blankets, and a tritium factory [27]. CFETR also requires further investigation to find a diverter that has a long service life, is cost efficient, and is compatible with remote maintenance [32].

In 1998, the Chinese government approved a national mega-project of scientific research initially known as the HT-7U superconducting tokamak and later renamed as the Experimental Advanced Superconducting Tokamak (EAST) in 2003 [59]. Its primary objective is to comprehensively explore the physics and technologies involved in advanced tokamak operations, particularly focusing on the mechanisms for power and particle management in steady-state operations. The scientific and engineering goals of the EAST project are to investigate the physics challenges of advanced steady-state tokamak operations and to establish the technological foundation for full superconducting tokamaks [59]. This device has demonstrated high performance and long pulse operation, which can be a significant reference for future device development, including ITER, CFETR, and DEMO [8]. Similar to ITER, EAST is the first fully superconducting tokamak that applies advanced divertor configurations and heating schemes [60]. EAST has successfully achieved a novel steady-state H mode without experiencing edge-localized modes (ELMs), for an extended duration surpassing hundreds of energy confinement times [61]. The ELM is the magnetohydrodynamic (MHD) instability that causes periodic spikes of heat and particle fluxes in the plasma-facing components [61]. This breakthrough solved one of the primary challenges confronted by ITER due to the basic long-pulse high-confinement operation (H mode scenario). Figure 17 illustrates the fusion roadmap proposed for the development of magnetic confinement fusion (MCF) in China towards fusion commercialization [27]. The roadmap includes both short- and long-term strategies, and one of the short-term strategies was the establishment of advanced platforms for research in plasma physics such as EAST [27].

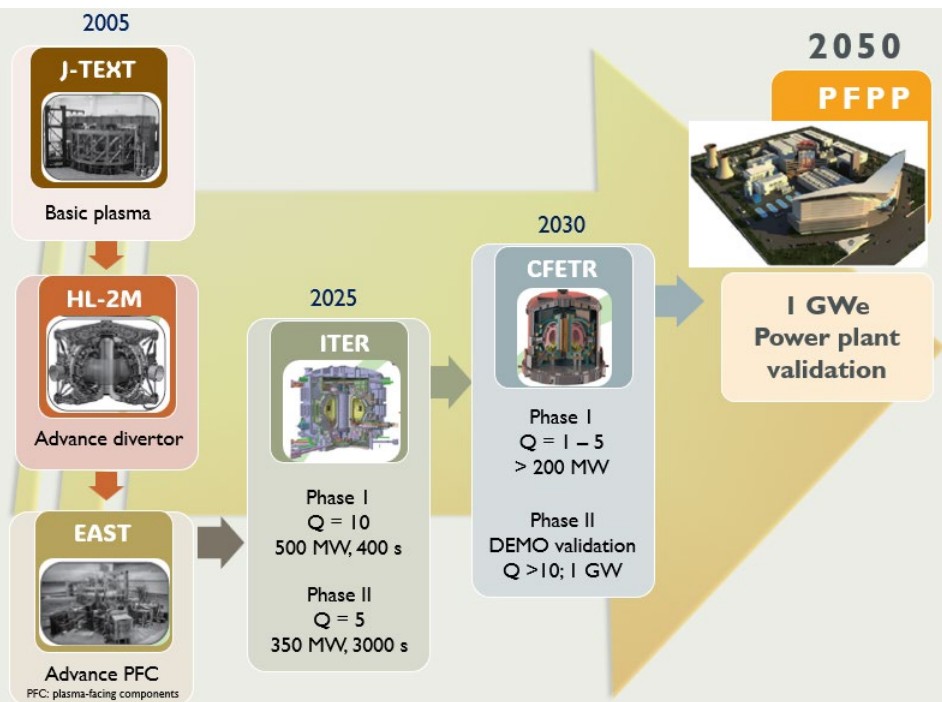

**Figure 17.** Roadmap for the development of MCF fusion devices in China.

China aims to achieve 50–200 MW fusion power with a $Q_{plasma}$ of 1–5, self-sufficiency with a tritium breeding ratio (TBR) of more than 1, and neutron radiation effects of 10 dpa during the first phase operation of CFETR. In the second phase, the fusion power is expected to increase by 1 GW with a $Q_{plasma}$ of more than 10 and a neutron irradiation effect of 50 dpa [32]. The phase transition requires alteration to the major and minor radius of the tokamak from 5.7 m and 7.2 m, respectively, to 1.6 m and 2.2 m, respectively. This DEMO device will highlight the stability control of combusted plasma, continuous power generation, fusion energy, and self-sustainability of tritium [32].

China also participates in the development of ITER and is responsible for providing several parts of the ITER tokamak, such as 31 units of magnet feeders, 18 superconducting correction coils, and electrical conversion components [62]. Four units of converter transformers, weighing 128 ton each, were successfully supplied to ITER in October 2022. China also supplied its pulsed power electrical system (PPEN) voltage transformers to ITER in 2023, which enabled the ITER tokamak to deliver the power into its magnet coils, heating, and current drive system during plasma pulses [62]. Over 70% of the technology gained from knowledge and experience in ITER will be applied to the construction of CFETR. In addition, more than 50% of CN-ITER-PA has been completed through qualified components received by ITER-IO and there is significant progress on ITER-PA [63]. Overall, ITER has been a great platform to accelerate the progress of MCF development in China since its participation in 2006.

### 3.7. Japan

Japan founded its Japan National Policy in 2021 to further accelerate the country's development in fusion science [29]. Figure 18 explains the detailed strategies of fusion science allocated in the Japan National Policy 2021. The strategies focus on magnifying the R&D promotion, demonstration, and utilization of fusion sciences and technology toward cleaner energy through domestic and international platforms. This strategy includes the development of fusion devices and visualization for commercialization. To date, Japan has recorded the highest number of tokamak devices available and planned for construction in the world. The country has 13 tokamaks, and 9 of them are currently operating, as listed in Table 7.

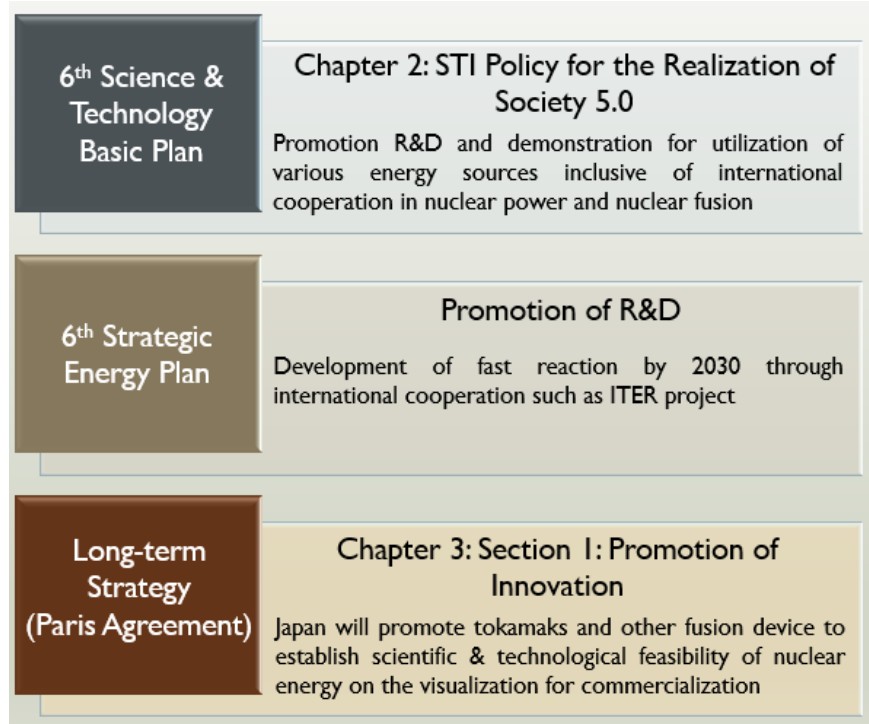

**Figure 18.** Fusion science in the National Japan Policy roll out in 2021.

**Table 7.** The operational tokamak fusion devices available at an experimental scale in Japan.

| Tokamak Type | Device Name | Organization |
| --- | --- | --- |
| Conventional | PLATO | Kyushu University |
| | HYBTOK-II | Nagoya University |
| | TOKASTAR-2 | Nagoya University |
| | JT-60SA | National Institutes for Quantum and Radiology Science & Technology |
| | PHiX | Tokyo Institute of Technology |
| Spherical | LATE | Kyoto University |
| | QUEST | Kyushu University |
| | TS-4U | |
| | TS-6 | |
| | TST-2 | The University of Tokyo |
| | UTST | |
| | HIST | University of Hyogo |

Japan is currently in the stage of completing the conceptual design of a fusion demonstration device (DEMO) known as JA-DEMO, in parallel to the construction of ITER. The deuterium (D) and tritium (T) fuels in JA-DEMO are supplied using gas puff, pellet injection, and neutral beam injection and require a closed D-T cycle for fuel recycling since the fuel combustion rate is around 1.7% [64]. JA-DEMO aims to demonstrate reliable power-plant-scale power generation, possess a self-sufficient supply of tritium, and have the potential to extend for commercialization [65]. The device will have a larger magnetic field and toroidal field than ITER. The larger size may introduce technical challenges to obtain stable electric power generation as low as 1.5 GW. Thus, the device will require performance testing once it transits into the engineering design phase before the completion of conceptual design in 2025 [65].

JA-DEMO is designed with major radius of 8.5 m and with a fusion power of 1.5 to 2 GW. Small fusion power allows for greater realistic design of the divertor for heat removal. Major changes in the design developed in the past decade have been observed since Japan changed its policy for the lower power target of JA-DEMO [66]. For instance, the internal structure and pressure tightness of the breeding blanket (BB) must be simplified and the size of radioactive waste facilities needs to be redesigned with a down-sizing factor or introducing temporary waste management facilities [66].

Figure 19 describes the roadmap of fusion science in Japan and its strategy towards DEMO construction for fusion power generation. The conceptual design of JA-DEMO is expected to be completed by 2025, and the next phase will progress into engineering design together with full-scale technology development [65]. Japan has initiated several programs in parallel with ITER to support the success of JA-DEMO, such as testing devices JT-60SA, fusion neutron source generation, and blanket development. The roadmap also includes additional research on large helical devices, high-power lasers, as well as social relation activities. All designs, testing, analysis, and verifications will validate the steady-state operation of JA-DEMO and its construction. For instance, JT-60SA is utilized to achieve the technical targets for ITER and realize DEMO by conducting the burn control and engineering test [67].

The demonstration of high-energy neutron irradiation is also crucial before DEMO design and fabrication. Thus, Japan has developed the neutron sources and facilities for post-irradiation examination and fusion neutron source (FNS), as a part of Broader Approach (BA) activities with Europe to verify a high-energy neutron irradiation environment operation. The verification testing is expected to be performed by 2030 and will provide the information on the initial irradiation data as a guidance to DEMO construction [67].

Broader Approach (BA) is one of the strategies by Japan for the realization of DEMO and supporting ITER. Some of the BA activities conducted jointly through international collaboration with Europe (EU) are described in Table 8 [67].

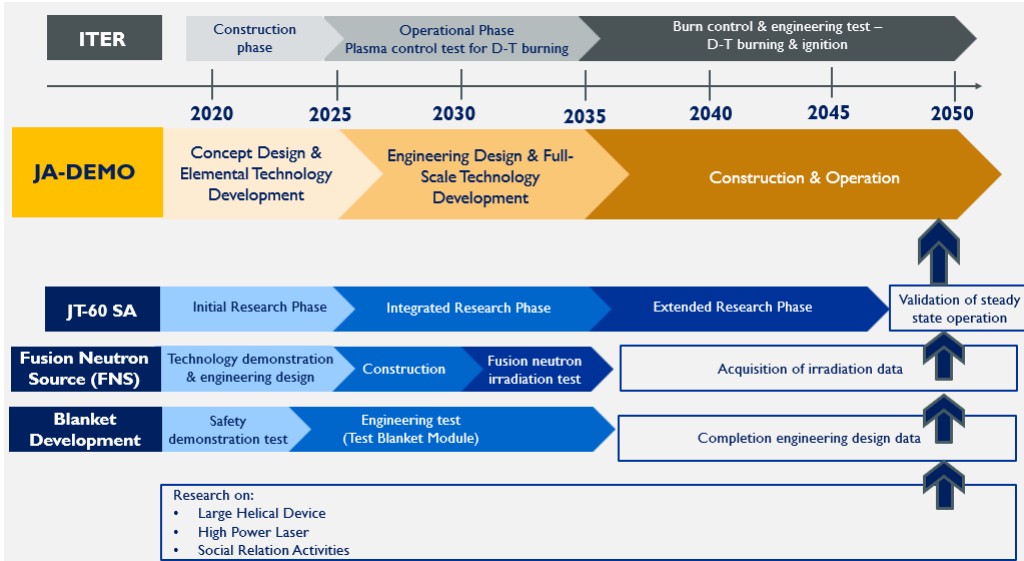

**Figure 19.** The strategy and road map approach of fusion energy in Japan in line with ITER development.

**Table 8.** Broader Approach (BA) activities jointly conducted by Japan and Europe (EU).

| Broader Approach (BA) | Aim |
|---|---|
| International Fusion Energy Research Center (IFERC) | DEMO design activities and R&D, computational simulation, and preparation for remote experiment |
| International Fusion Materials Irradiation Facility (IFMIF) | Sophistication of the prototype accelerator toward long-term continuous operation, and concept design of fusion neutron sources based on past activities |
| JT-60SA | Development of operation scenarios for ITER and DEMO and enhancement of devices |

Japan is one of the ITER members and the country is responsible for providing nine units of toroidal field coil magnets with cases and superconductors for the central solenoid, as well as deliveries of a neutral beam system and construction of the neutral beam test facility in Padua, Italy [68]. Mitsubishi Heavy Industries Ltd./Mitsubishi Electric Co. managed to fabricate and assemble the first toroidal field coil magnets and cases in 2017, whereas the second units were achieved by Keihin Product Operations/Toshiba Corp. Furthermore, Japan successfully shipped the final superconductor materials to the US in 2023 [68]. The materials are crucial parts of the modular coils that complete the central solenoid magnet of the ITER tokamaks. Japan also completely delivered the power supply components to the neutral beam test facility in November 2023, which is one of the major milestone achievements for the project. ITER is considered a core program in the Japan road map for DEMO establishment, where its scientific and technology feasibility is demonstrated based on the method to control the extended burning plasma, the vitality of reactor technologies in an integrated system, and performance of the device's blanket [69].

### 3.8. Russia

Russia is actively exploring a pathway towards controlled fusion (CF) through magnetic confinement, in conjunction with fission power, within the framework of the SC Rosatom's State Program. The CF is intertwined with the development of tokamak devices and their enabling technologies such as T15MD and Globus M2, along with active par-

ticipation in ITER. Russia firmly believes that fusion–fission hybrid systems (FFHSs) are indispensable for advancing both fusion and fission power engineering, which is regarded as a significant component in global nuclear energy development within Russia [70]. A divertor tokamak T-15MD equipped with copper coils is being used for the FFHS and the project is forecasted to achieve full-scale operation with a heating power capacity up to 20 MW by 2024 [70]. This developmental trajectory aligns with contemporary CF research objectives.

Domestic R&D projects and pre-conceptual designs oriented towards DEMO-RF in Russia have also been initiated at the National Research Centre Kurchatov Institute and the construction of this facility is anticipated to take place after 2055 [70]. The primary objectives of this DEMO design and its corresponding R&D encompass selecting fundamental fusion technologies that are suitable for the tokamak to generate electric power up to the gigawatt level. DEMO-RF represents a DEMO concept founded on traditional tokamak design and is currently under development by a Russian consortium in the Russian Federation. The construction timeline for DEMO-RF is projected for completion by 2055 and it aims to demonstrate a net engineering gain ($Q_{eng}$) of higher than 1 [10]. The current conceptual design for DEMO-RF envisions its utilization either as a standalone fusion energy system or as a fusion–fission hybrid facility, featuring high-temperature superconducting magnets capable of generating a total magnetic field exceeding 8 Tesla and sustaining a plasma current of approximately 5 Mega Amperes [10].

As an initial milestone, DEMO-RF is designated as a DEMO fusion neutron source (DEMO–FNS), and it is expected to have the sustainable operation of DT-fusion power up to 40 MW and fission power of 400 MW [70]. This reactor is scheduled for design and construction by 2033, to not only produce energy from fusion but also leverage fusion-generated neutrons to convert non-fissile uranium into fissile nuclear material or to eliminate long-lived radioactive waste. The DEMO-FNS is envisioned as a relatively compact fusion facility that operates in a steady-state mode [71]. Its initiative is integral to the Russian Federation, which accelerated the pathway toward a fusion power plant by 2050 [10]. The pilot hybrid facility construction is then envisioned by 2045, as well as the fusion power plant to surpass current scales by 2055, and the establishment of a commercial hybrid plant also by 2055 [70]. Figure 20 illustrates the milestones of each facility with the respective R&D charted through the Rosatom Fusion Strategy [70].

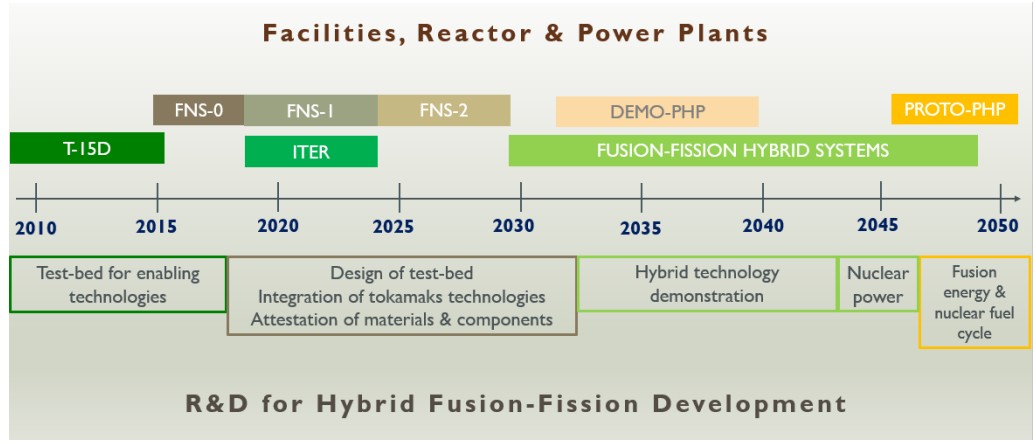

**Figure 20.** Milestones of the Rosatom Fusion Strategy.

The construction of the pilot hybrid plant (PHP) is scheduled for completion by 2030. The design objectives include achieving 40 MW of fusion power, 500 MW of total thermal power, and 200 MW of electric power to attain an engineering $Q_{eng}$ of approximately 1 [72]. By 2040, plans are in place for the construction of the industrial hybrid plant. Collaborative efforts are underway for the engineering design of the DEMO-FNS device, intended to demonstrate hybrid and molten salt technologies by involving partnerships with Rosatom

public organizations and universities. Progressing with the ITER project, PHP will potentially expedite the realization of DEMO-RF and contribute to the establishment of a commercial fusion power plant in Russia by 2050 [72].

Russia plays a vital role in the ITER project, including managing the funding for the ITER organization, producing and supplying 25 pieces of advanced high-end equipment, and contributing Russian personnel to aid in the completion of the ITER project [73]. Figure 21 describes the main responsibilities of Russia within the ITER project [74]. The Troitsky Institute for Innovation and Fusion Research is the primary hub for the project's R&D in Russia, fundamentally in the related plasma and laser research [73].

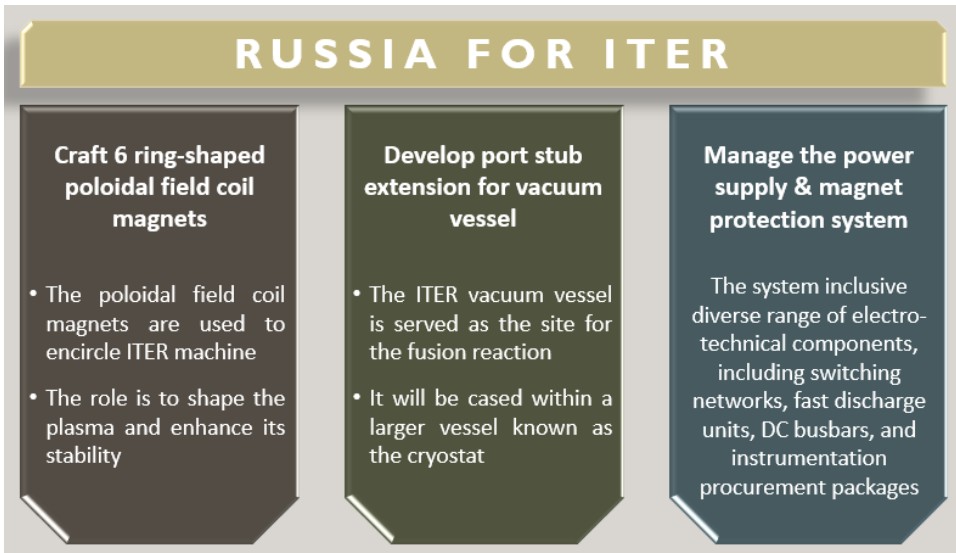

**Figure 21.** Responsibilities encompassed by Russia for ITER's development.

Russia also has developed a versatile mid-sized tokamak device, named T-15MD, to address the critical challenges that have been encountered in ITER. It is an upgraded Russian tokamak and aims to expand the operational capabilities of "ITER-complementary" machines by focusing on determining the optimal operating parameters for ITER and future fusion reactors [75]. The device successfully achieved its inaugural stable plasma operation at the Kurchatov Institute in the Russian Federation in April 2023. It is capable of generating a toroidal magnetic field of 2 Tesla with additional heating systems and delivering a cumulative power input of up to 20 MW [10]. The targeted plasma current of T-15MD is 2.0 Mega Amperes within 10 s [10]. Spanning a decade of construction, the T-15MD tokamak's experimental program has had a significant contribution to ITER's operational objectives and the development of future power generation facilities. The establishment of future fusion power generation in Russia is also supported by other experimental scale devices that currently operating. These tokamak devices are listed in Table 9.

**Table 9.** Tokamaks operating at an experimental scale in Russia.

| Tokamaks Type | Device Name | Organization |
|---|---|---|
| Conventional | FT-2 | Ioffe Institute |
| | TUMAN-3M | |
| | T-15MD | National Research Centre Kurchatov Institute |
| | T-11M | Troitsk Institute for Innovation and Fusion |
| Spherical | Globus-M2 | Ioffe Institute |
| | GUTTA | Saint Petersburg State University |

## 4. Operational and Commercialization Challenges for Fusion Development

Through progressive experimental works and design configurations towards ITER and DEMO construction, a number of challenges and setbacks in many aspects related to technology, national and international policy, and financial are being learned and overcome towards successful device integration into the grid. Figure 22 summarizes the overall technical challenges experienced while developing and transforming fusion devices and facilities on a large scale [10].

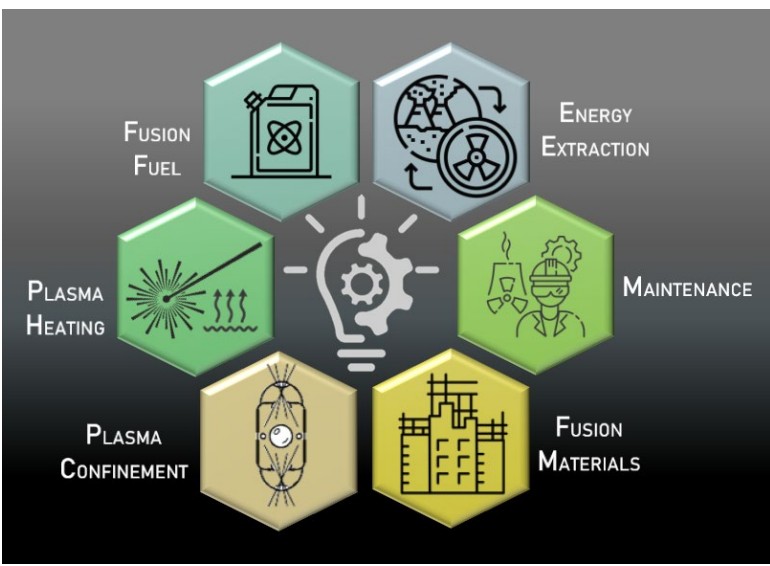

**Figure 22.** Overall technical challenges of establishing fusion power plants for commercialization.

### 4.1. Plasma Heating and Confinement

Plasma heating and plasma confinement are among the major technical challenges that have been observed while designing the fusion devices for the experimental, demo, and commercialization scales [10]. Fusion requires a stable and extremely hot plasma to be initiated. Among the techniques to introduce the extra energy into the fusion device for the hot plasma generation are neutral beam injection and high-frequency electromagnetic radiation. Neutral beam injection involves propelling high-speed neutral particles into the plasma to be ionized and impart the energy through collisions. The fusion materials must be kept away from the wall since any melting or evaporation at the wall surface will release impurities in the fusion material, potentially diminishing fusion power through radiative cooling and fuel dilution [76]. Utilizing strong magnetic fields will help to maintain a safe distance between the plasma and the wall, by encasing the hot plasma within a torus-shaped reaction chamber and preventing it from approaching the reactor's inner wall [76]. However, building bulky-size devices may impact the cost and require larger space for construction. Thus, the challenge includes developing the techniques to eliminate impurities and reaction byproducts from the plasma as well as choosing suitable material for the wall reactor.

Confining high-temperature plasma for a sufficient duration to allow the nuclei to fuse also can be a challenging accomplishment. The design construction must be able to achieve elevated plasma temperature and densities while maintaining plasma stability and confinement [77]. Achieving perfect plasma confinement is tricky due to the turbulent stream of plasma that can transport heat and particles to the wall. Additionally, any periodic instabilities may result in bursts of hot plasma reaching the wall and ultimately causing damage that reduces the lifespan of the device [77]. These issues raise concern for the safety and operational robustness of the device. Large-scale superconducting magnets are capable of tackling the challenges of higher temperature requirements. However, the high cost, bulky size, and energy intensiveness are unfavorable for building and running this

extensive superconducting electromagnet, although it is necessary for generating magnetic fields to confine the plasma.

### 4.2. Fusion Fuel Sustainability and Energy Extraction

The sustainability of fusion fuels such as deuterium and tritium is concerning in establishing commercial fusion power plants in the country. While deuterium can be extracted from seawater, tritium must be generated or "bred" within the fusion facility itself. Tritium is a radioactive, heavy hydrogen isotope with a half-life of 12.3 years that only occurs naturally in small quantities in the upper atmosphere. Tritium can be bred through a nuclear reaction between fusion-produced neutrons using $^6Li$ and $^7Li$, which involves enveloping the reactor with a "blanket" to compose the materials [9]. The neutron capture reaction using $^6Li$ also yields energy and contributes to heat output in the reactor. Figure 23 illustrates the fusion power generation and fusion reaction rate according to the fraction of tritium used in the fuel and fusion fuel reactions. One atomic percent of tritium can generate 44 MW of fusion power provided the fusion rate is 1.3 and $1.1 \times 10^{19}$/s through D-T and D-D reactions, respectively. By multiplying with 50 atomic percent of tritium, the generated fusion power is expected to be approximately over 50-fold higher [78]. A fusion facility must be able to estimate the amount of fusion fuel required to ensure the generated fusion power is manageable and under control for efficient energy extraction. A self-sustainable tritium power plant with a complete fuel cycle is the preferable option to build an operable fusion facility in addition to complex design, detailed safety considerations, and maintenance.

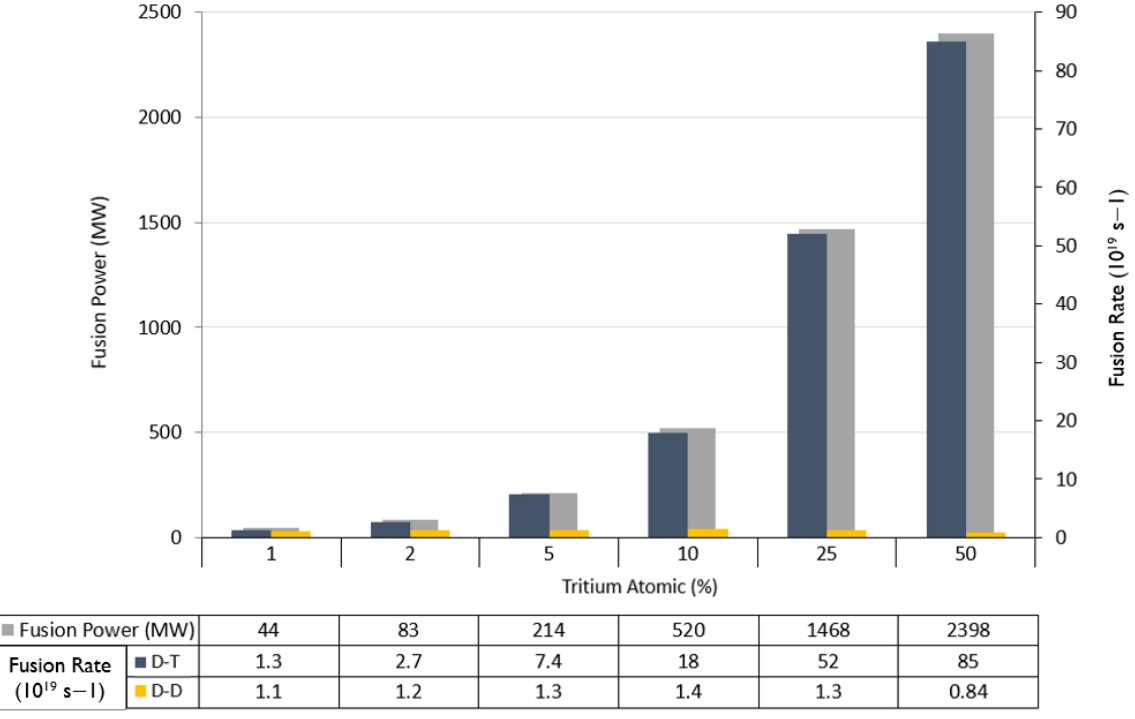

| ▨ Fusion Power (MW) | | 44 | 83 | 214 | 520 | 1468 | 2398 |
|---|---|---|---|---|---|---|---|
| **Fusion Rate** ($10^{19}$ s−1) | ■ D-T | 1.3 | 2.7 | 7.4 | 18 | 52 | 85 |
| | ■ D-D | 1.1 | 1.2 | 1.3 | 1.4 | 1.3 | 0.84 |

**Figure 23.** Estimation of generated fusion power and fusion rate according to different tritium atomic fractions.

Currently, comprehensive design and verified solutions for tritium breeders, associated fuel loops, plant integration, and handling facilities for tritium inventory are still limited worldwide. ITER has included the evaluation of various tritium breeder blanket modules that encircle the reactor vessel as one of its main objectives, which will ultimately demonstrate the technical feasibility of tritium fuel breeding [31]. The tritium inventory within the fuel cycle can be intricately linked to the demands of pellet manufacturing and hydrogen isotope separation [64]. Table 10 summarizes the approximate amount of

tritium inventory in the plants that operated based on the D-T reaction and their utilization according to the fusion facilities [71]. The estimation indicates DEMO will require at least double the fuel inventory compared to ITER and will have an inlet fuel flow of almost 1000 g/h. Thus, the sustainability of the fuel supply and the capabilities for hydrogen isotope separation must be considered while developing facilities for commercial power plant operation.

**Table 10.** Fuel inventory and fuel cycle of tritium in different D-T fusion facilities [64].

| Fusion Facility | Fuel Inventory (g) | Inlet Fuel Flow |
|:---:|:---:|:---:|
| ITER | $\leq$4000 | ~300–1000 g/h |
| DEMO | $\leq$10,000 | ~1000 g/h |
| DEMO-FNS | $\leq$2000 | ~200 g/h |
| CFETR | $\leq$4000 | ~300 g/h |
| JET DTE1 | 20 | 100 g/all time |
| JET DTE2 | 60 | $\leq$135 g/day |
| TFTR | $\leq$25 | 90 g/all time |

*4.3. Technical and Engineering Limitation of Nuclear Fusion Devices and Facility*

Developing a successful magnetic fusion device for energy production is challenging, particularly regarding its performance during abnormal events like plasma disruptions and edge-localized modes (ELMs), which cause sudden energy releases and high transient power loads on reactor surfaces [79]. Maintaining reactor integrity during these events is crucial. Plasma instabilities, especially during transitions between confinement regimes, will activate ELMs and heat the plasma-facing components (PFCs). As fusion device size and power increase, new challenges emerge, including plasma–material interactions and PFC performance during abnormal events. Increasing the "wetting area" on divertor plates can mitigate heat loads and prevent PFC damage, which can be achieved through divertor optimization or buffer zone implementation [79]. The divertor detached regime involves injecting neutrals into the divertor space to enhance the energy dissipation above PFC surfaces, enlarging the "wetting area". However, using low-Z materials as buffer zones may affect core stability and cause temperature drops. While the detached divertor can mitigate small ELMs, large ELMs pose a risk to divertor plates and the adjacent components [79]. Addressing this requires optimizing divertor design and detachment methods. Accurately predicting heat and particle loads is crucial for selecting optimal PFC materials. The loss of core plasma confinement also poses concerns for divertor plate lifetimes and core plasma contamination.

Tokamak ITER also experienced challenges on the heat exhaust that have been addressed by employing high-heat flux components based on tungsten monoblock technology and partially detached divertor operation [80]. However, to ensure this solution's applicability to DEMO, investigations into advanced divertor configurations and the use of liquid metals as plasma-facing materials are underway, with a dedicated DTT facility being constructed. Qualifying structural materials capable of withstanding intense neutron flux and possessing benign activation properties are essential. Materials like EUROFER have been produced and could be used initially in DEMO operation, where nuclear damage is expected to be below 20 dpa in the first phase [80]. However, for the second phase, new materials need qualification, necessitating facilities like IFMIF with a neutron energy spectrum simulating that of a fusion reactor [80].

Furthermore, hydrogen isotopes can exist in various isotopic forms called isotopologues, each with similar but not identical physical properties and reaction kinetics. Understanding their behavior is crucial, as it can be both beneficial and detrimental for the fuel cycle in a fusion facility. Unlike hydrogen permeability, tritium transport and retention are not well understood, making quantification and modeling challenging without sufficient

experimental data [81]. A fusion fuel cycle involves continuously removing, processing, and reinjecting fuel to the plasma while safely containing these fuels. The leading method is in situ tritium breeding for prompt extraction and processing to overcome losses through decay, permeation, and retention [81]. This requires processes capable of separating specific isotopes, often involving complex techniques like cryogenic distillation. Impurities must be removed cyclically to prevent accumulation and recover tritium, which can originate from various sources such as system leakages, material outgassing, or plasma control injections into the tokamak [81]. These challenges underscore the complexity of fusion fuel cycles and the need for advanced separation technologies.

### 4.4. Maintenance and Safety Considerations

It is also crucial for commercial fusion power plants to have a comprehensive maintenance system to secure the plants' safety, which includes the plant operational concept of "gate-to-cradle". Taylor et al. [38] reported that two critical safety elements identified for ITER are the confinement of radioactive materials and the exposure limit to radiation, which exposed the facility to several risks including fire, explosion, earthquake, and the cut-off of electric power. Understanding the fusion reaction able to provide better safety measures and design of the facility and device. Taylor et al. [38] summarized four characteristics that are essential for safety performance and low environmental impact, which are as follows:

- The impact of normal operation: minimize releases of radioactive and other potentially hazardous substances, in gaseous, liquid, or aerosol form to the environment.
- The potential accidents initiated by internal faults in the facility or external events: minimize the frequency of plant failures that could initiate an accident sequence by using a strong safety design.
- The well-being of personnel: ensure radiation exposure is reasonably low and minimize other occupational hazards.
- Consideration of radioactive waste in solid form: ensure the quantity/level of activation/contamination is minimal and ensure a safe and secure route of disposal.

A fusion facility must aim to operate without posing any public health or environmental risks by including radioactivity safety and environmental impact in its operational analysis. Thus, the primary consideration when developing fusion facilities must include material handling, material activation, and site selection [82]. One way that can allow the handling of nuclear materials including waste is through careful evaluation of the materials' response to the environment as well as to the handlers. For instance, additional attention must be given when handling tritiated waste and unique materials such as beryllium [83]. The containment of the building surrounding the reactor is essential to eliminate the necessity for emergency plans involving population evacuation or sheltering [82]. Thus, the challenges are around establishing a safe working culture, device design, and facility operation.

### 4.5. Private Funding Opportunity

The progress and enhancements in the design and construction of future reactors demand sufficient funding to realize cost-effective, safe, and dependable fusion energy production. Up to 2023, the fund for the global fusion program had gained a total of USD 6.2 billion, where USD 5.9 billion is contributed by private companies [19]. The US experienced a noticeably increased transition of participation by private sectors into fusion R&D. The private funding in the past years has been increasingly channeled into fusion companies located in Canada, the UK, the EU, China, Japan, and Australia [84]. Globally, there are an estimated 50 private companies operating in fusion industries as of 2023, where rapid blooming was observed within between the years 2017 and 2021, increasing from 15 to 45 companies within 4 years [19]. Figure 24 illustrates the dynamic involvement of private companies in developing the fusion technology around the world (FIA, The Global Fusion Industry in 2023).

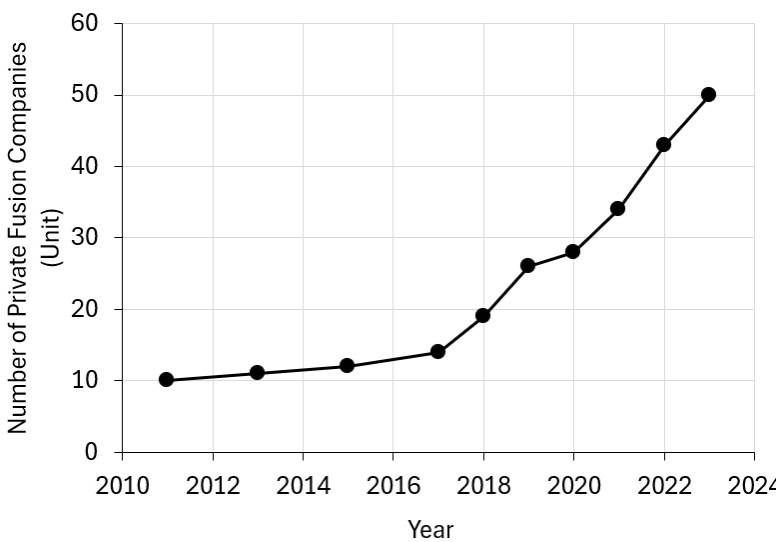

**Figure 24.** Total number of private companies involved in fusion worldwide.

Private companies can utilize their expertise in scientific and technological fields like plasma physics, material science, and nuclear engineering to explore markets beyond electricity generation, such as medical technology, robotics, and superconducting technology. This diversification enables them to generate short-term profits while working toward long-term goals. Spin-offs allow for the commercial use of intellectual property rights from research and development efforts, providing additional revenue streams for private companies and fostering collaboration between research institutions and industry [85]. Governmental support, especially through energy policies, plays a crucial role in sectors with significant time and capital requirements. Government involvement helps mitigate the risks associated with private investments by establishing regulatory frameworks for fusion reactor construction and operation [85]. Engaging with the public and policymakers is vital for technology acceptance, necessitating effective communication of safety-related topics and providing essential information to decision makers [85]. Table 11 lists the project background of some private companies in the world that have invested more than USD 200 million in fusion research and development works.

**Table 11.** Project description of the private companies with a total investment larger than USD 200 million [19].

| Company | Total Declared Funding ($) | Project Description |
|---|---|---|
| **Zap Energy** | 208,000,000 | Zap Energy is building a low-cost, compact, and scalable fusion energy platform that confines and compresses plasma without magnetic coils or high-power lasers. Zap's quickly advancing sheared-flow-stabilized Z-pinch technology provides compelling fusion economics and requires orders of magnitude less capital than conventional approaches |
| **SHINE Technologies** | 700,000,000 | SHINE is commercializing and industrializing near-term applications of fusion, like inspecting industrial components through neutron imaging and producing medical isotopes. These applications create tremendous social and economic value and allow us to build and practice the capabilities we believe are essential for deploying fusion energy to billions of people |
| **Tokamak Energy** | 250,000,000 | Tokamak Energy is the only private fusion company to have more than 10 years of experience of designing, building, and operating tokamaks. It is focused on developing fusion pilot plants for the 2030s using spherical tokamaks and high temperature superconducting (HTS) magnets, as well as developing its HTS magnet technology for other industry applications |

**Table 11.** *Cont.*

| Company | Total Declared Funding ($) | Project Description |
|---|---|---|
| **Helion** | 577,000,000 | Building the world's first fusion power plant to enable a future with unlimited clean electricity |
| **General Fusion** | >300,000,000 | General Fusion is pursuing a fast, practical path to bring fusion power to the market by the 2030s using its proprietary magnetized target fusion (MTF) technology |
| **ENN Science And Technology Development Co., Ltd.** | 400,000,000 | ENN is committed to generating fusion energy in an environmentally friendly and cost-effective manner. A number of devices are being designed and built to support our vision for commercial ST p-11B fusion |
| **Commonwealth Fusion Systems** | >2,000,000,000 | Commonwealth Fusion System's (CFS) mission is to deploy fusion power plants to meet increased global energy demand and decarbonization goals as fast as possible. CFS leverages decades of research in tokamaks combined with new groundbreaking high-temperature superconducting (HTS) magnet technology. CFS is currently constructing SPARC, a Q~10 demonstration plant based on peer-reviewed science, using fusion fuels. |
| **TAE Technologies** | >1,200,000,000 | TAE Technologies (pronounced T-A-E) is developing safe, non-radioactive, cost-effective, commercial fusion energy capable of sustaining the planet for centuries. Through its unique approach to fusion, TAE has developed spinoff applications in life sciences, energy storage, electric mobility, and fast charging to create a complete clean energy ecosystem. Multidisciplinary and mission driven by nature, TAE is leveraging proprietary science and engineering to create a bright future. |

Confidence in technology capability, sustainability, and intellectual property are among the critical elements that influence the trust and interest of investors and guarantor sentiments [50]. The Fusion Industry Association (FIA) reported a survey conducted among the private sector to identify the major and minor challenges for fusion development in the current scenario until 2030. The results indicated the aim to achieve the high gain of fusion power (Q) and obtaining funds are the major challenges, whereas having cryo-plants for heat management, plasma exhaust system, and geo-politics are regarded as the most minor challenges for fusion development [19]. The details of the survey results are summarized in Figure 25.

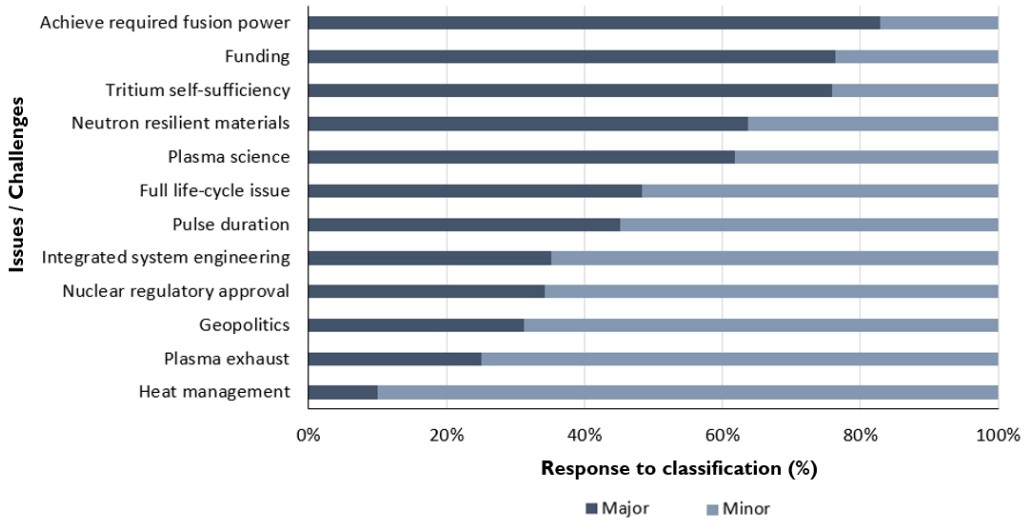

**Figure 25.** Response from industry players in fusion for the classification (major/minor) of related challenges and issues.

## 5. Conclusions

Meeting the environmental targets and energy security goals by 2050, or earlier, demands innovative design approaches, collaboration, and consensus strategy building to accelerate and increase the spread of the adoption of nuclear fusion power. Many proposed designs for grid-ready fusion reactors are progressing at the experimental scale and require major technical designs and sufficient sustainable funding to surpass the prototype stage fusion device, as well as for the planned DEMO to be translated within the timeline. The timeframe for the successful deployment of fusion energy will depend on the mobilization of resources through global partnerships and collaboration, as well as the industry's ability to develop, validate, and qualify the emerging fusion technologies. In parallel, it is imperative to address the development of the essential nuclear infrastructure, including defining requirements, standards, and best practices for realizing fusion as a clean and secure future energy source.

**Author Contributions:** Writing—original draft preparation, M.M.; writing—review and editing, N.D.Z. and T.N.A.T.H.; conceptualization and review, A.M.S. and S.S.M.L.; supervision, A.M.S. and S.S.M.L. All authors have read and agreed to the published version of the manuscript.

**Funding:** Financial support was from the MRA-FP project (Grant Number: 015MD0-139).

**Institutional Review Board Statement:** Not applicable.

**Informed Consent Statement:** Not applicable.

**Data Availability Statement:** Not applicable.

**Acknowledgments:** The authors would like to acknowledge the Universiti Teknologi PETRONAS and MRA-FP project.

**Conflicts of Interest:** The authors declare no conflicts of interest.

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
