# Peer review of "Global Development and Readiness of Nuclear Fusion Technology as the Alternative Source for Clean Energy Supply"

_sustainability, doi:10.3390/su16104089_

Round 1

Reviewer 1 Report

Comments and Suggestions for Authors
  1. he review paper explores the potential of nuclear fusion technology in achieving sustainable energy goals and mitigating climate change. It analyzes the global overview and development of nuclear fusion devices, indicating progress in the experimental phase but challenges in commercial applications. Additionally, it underscores the significance of safety and environmental impacts of fusion equipment and the growing private sector involvement in fusion research and development.
  2.  
  3. 1. Condensing the summary section, such as presenting the ITER programme more succinctly.

  4.  
  5. 2 Clarifying Figure 8 within the "SPIN-OFF MARKET" section to enhance comprehension, perhaps by reformatting the image for better comparison.

  6.  
  7. 3 Adding units to graphs, like in Figure 11, for clarity and interpretation.

  8.  
  9. 4 Segmenting the challenges facing fusion development and commercialization for improved organization.

  10.  
  11. 5 Addressing the limited discussion of specific technical details and engineering implementation aspects of nuclear fusion equipment by providing deeper technical analyses.

  12.  
  13. 6 Exploring the specifics and dynamics of private sector involvement in fusion research and development in greater depth.

Comments on the Quality of English Language

it is ok

Reviewer 2 Report

Comments and Suggestions for Authors

The review discusses fusion development in the EU and 5 major countries such as US, UK, China, Japan, and Russia.  Meanwhile the article provides an overview on the utilization of nuclear energy as a clean energy source, as well as the strategies and progress towards establishing successful commercial fusion energy to the grid and transition to a reliable clean energy source. Finally, significant challenges were identified from the perspective of device efficiency and robustness, sustainable funding, facility maintenance and safety. The manuscript can be accepted after addressing the below comments to enhance recent statistical data.

Abbreviations such as ITER.... must be defined at first use.

Add the recent report of "Energy Information Administration (EIA) reported the share of total world electricity generation by nuclear" for example: 2023.

Similarly, the Fig 2, 3 needs to be updated until the recent years.

Reviewer 3 Report

Comments and Suggestions for Authors

The manuscript "Global Development and Readiness of Nuclear Fusion Technology as The Alternative Source for Clean Energy Supply" provides an overview on the utilization of nuclear energy as a clean energy source, as well as the strategies and progress towards establishing successful commercial fusion energy to the grid and transition to a reliable clean energy source. This information will be very useful.

The overview focuses on the fusion development in the 5 major countries which are US, UK, China, Japan, and Russia. However, it would be convenient that in the first approach to the study the authors mention which are all the countries in the world that have this technology, and then refer to the 5 countries that they have selected. Please incorporate this information.

The manuscript mentions very little about standards. This should be included.

Reviewer 4 Report

Comments and Suggestions for Authors

The authors provide an overview of the utilization of nuclear energy, as well as the strategies and progress towards establishing successful commercial fusion energy to the grid and transition to a reliable clean energy source. This interesting overview presents various approaches and technologies implemented in specified countries. However, I have also noticed a few minor issues that should be corrected.

lines 22-23: "The overview focuses on the fusion development in the EU and 5 major countries which are US, UK, China, Japan, and Russia" - in the section where detailed information about specific countries is delivered there is no information about the UK described in a similar way to other countries.

figure 8: how to interpret the values such as 33,11, etc. in the pie chart or 6,7, etc. in the lower chart. Why two different types of charts?

line 397: "is planned to begin by the middle of 2021" Has this plan been implemented?

line 451: "[51] estimated more fusion power" - the sentence should start with a correct subject (not a number which is a reference to literature)
line 735: "[71] summarized four characteristics" - likewise

Some references are not properly described, for example [24] - it is only the name “DOE Explains Tokamaks” (the same for [33]) or [25] "L. Zabeo et al." the full list of the author's names should be delivered (the same comment applies to many other items), the link for [20] is incorrect.
